# TIAM-1/GEF can shape somatosensory dendrites independently of its GEF activity by regulating F-actin localization

Leo TH Tang[1†], Carlos A Diaz-Balzac[1†], Maisha Rahman[2], Nelson J Ramirez-Suarez[1], Yehuda Salzberg[1‡], Maria I Lázaro-Peña[1], Hannes E Bülow[1*]

[1]Department of Genetics, Albert Einstein College of Medicine, Bronx, United States; [2]Dominick P. Purpura Department of Neuroscience, Albert Einstein College of Medicine, Bronx, United States

**Abstract** Dendritic arbors are crucial for nervous system assembly, but the intracellular mechanisms that govern their assembly remain incompletely understood. Here, we show that the dendrites of PVD neurons in *Caenorhabditis elegans* are patterned by distinct pathways downstream of the DMA-1 leucine-rich transmembrane (LRR-TM) receptor. DMA-1/LRR-TM interacts through a PDZ ligand motif with the guanine nucleotide exchange factor TIAM-1/GEF in a complex with *act-4/Actin* to pattern higher order 4° dendrite branches by localizing F-actin to the distal ends of developing dendrites. Surprisingly, TIAM-1/GEF appears to function independently of Rac1 guanine nucleotide exchange factor activity. A partially redundant pathway, dependent on HPO-30/Claudin, regulates formation of 2° and 3° branches, possibly by regulating membrane localization and trafficking of DMA-1/LRR-TM. Collectively, our experiments suggest that HPO-30/Claudin localizes the DMA-1/LRR-TM receptor on PVD dendrites, which in turn can control dendrite patterning by directly modulating F-actin dynamics through TIAM-1/GEF.
DOI: https://doi.org/10.7554/eLife.38949.001

*For correspondence:
hannes.buelow@einstein.yu.edu

†These authors contributed equally to this work

Present address: ‡Weizmann Institute, Rehovot, Israel

Competing interests: The authors declare that no competing interests exist.

## Introduction

Neurons are highly polarized cells, which comprise a single axon and often elaborately sculpted dendritic arbors. Dendrites receive input from other neurons or the environment, whereas the single axon transmits information to other neurons. The nervous system is formed by a myriad of specific synaptic connections between neurons and the formation of these connections is influenced by the shape and complexity of dendritic arbors. Both genes that act within the developing neurons and in surrounding tissues are crucial to establish distinct dendritic structures during development (*Jan and Jan, 2010*; *Dong et al., 2015*; *Lefebvre et al., 2015*). Of note, defects in dendrite morphology have been found in various neurological disorders (*Kaufmann and Moser, 2000*; *Kulkarni and Firestein, 2012*).

Both dendritic and axonal morphology is driven by the cytoskeleton and regulators of the cytoskeleton have consequently important functions in neuronal development. Major components of the cytoskeleton include actin and tubulin, which form filamentous polymers named F-actin and microtubules, respectively (*Brouhard, 2015*; *Pollard, 2016*). F-actin exists in unbranched and branched forms, whereas microtubules are generally unbranched. These filament-like polymers are not static, but highly dynamic structures due to the constant association and dissociation of monomers at either end. A plethora of proteins bind to and modulate polymerization and depolymerization of both F-actin and microtubules in neurons (reviewed in (*Dent et al., 2011*; *Kapitein and Hoogenraad, 2015*; *Konietzny et al., 2017*). F-actin and microtubules are important for countless aspects of

neuronal function and development in both axons and dendrites, including differentiation, migration and the elaboration of axonal and dendritic processes (*Jan and Jan, 2010*; *Dent et al., 2011*).

Regulation of the cytoskeleton is controlled by dedicated signaling pathways, which often originate with cell surface receptors. These receptors utilize regulatory proteins such as guanine nucleotide exchange factors (GEFs) or GTPase-activating proteins (GAPs), which, in turn, modulate the activity of small GTPases. For example, the RacGEF Tiam-1 regulates activity-dependent dendrite morphogenesis in vertebrates (*Tolias et al., 2005*), likely by activating the small GTPase Rac1. Activated GTPases such as Rac1 then bind the WASP family verprolin-homologous protein (WVE-1/WAVE) regulatory complex (WRC) of actin regulators to promote actin polymerization and branching (*Chen et al., 2010*).

The polymodal somatosensory neuron PVD in *Caenorhabditis elegans* has emerged as a paradigm to study dendrite development. The dendritic arbor of PVD neurons develops through successive orthogonal branching (*Oren-Suissa et al., 2010*; *Smith et al., 2010*; *Albeg et al., 2011*) (*Figure 1A*). During the late larval L2 stage primary (1°) branches first emerge both anteriorly and posteriorly of the cell body along the lateral nerve cord. In subsequent larval stages, secondary (2°) branches emanate orthogonally to bifurcate at the boundary between the lateral epidermis and muscle to form tertiary (3°) branches. These, in turn, form perpendicular quaternary (4°) branches to establish the candelabra-shaped dendritic arbors, which have also been called menorahs (*Oren-Suissa et al., 2010*). Previous studies have shown that an adhesion complex consisting of MNR-1/Menorin and SAX-7/L1CAM functions from the skin together with the muscle-derived chemokine LECT-2/Chondromodulin II to pattern PVD dendrites. This adhesion complex binds to and signals through the DMA-1/LRR-TM leucine rich transmembrane receptor expressed in PVD neurons (*Liu and Shen, 2011*; *Dong et al., 2013*; *Salzberg et al., 2013*; *Díaz-Balzac et al., 2016*; *Zou et al., 2016*). DMA-1/LRR-TM shows great similarity in domain architecture with the LRRTM family of leucine rich transmembrane receptors in humans (*Laurén et al., 2003*), but limited sequence homology (data not shown). The signaling mechanisms that operate downstream of the DMA-1/LRR-TM receptor in PVD dendrites have remained largely elusive.

Here, we show that DMA-1/LRR-TM forms a complex with the claudin-like molecule HPO-30 (*Smith et al., 2013*), which is required for surface localization and trafficking of the DMA-1/LLR-TM receptor in PVD dendrites. Consistent with these observations, DMA-1/LRR-TM functions genetically in the Menorin pathway together with *hpo-30/Claudin*, as well as the guanine nucleotide exchange factor (GEF) *tiam-1/GEF* and *act-4/Actin*. The signaling complex is required for both the correct localization of F-actin to, and the exclusion of microtubules from the distal endings of developing somatosensory dendrites. Intriguingly, TIAM-1/GEF appears to function independently of its Rac1 guanine nucleotide exchange factor activity. Biochemical experiments show that DMA-1/LRR-TM can form a complex with TIAM-1/GEF and ACT-4/Actin. Collectively, our experiments suggest that HPO-30/Claudin regulates surface expression of the DMA-1/LRR-TM receptor, which can modulate F-actin dynamics and localization through TIAM-1/GEF independently of GEF activity.

## Results

### Isolation of genes that function in PVD and FLP dendrite formation

From genetic screens for mutants with defects in PVD patterning (see Material and methods for details), we obtained recessive alleles of the leucine-rich repeat (LRR) single-pass transmembrane (TM) receptor *dma-1/LRR-TM* and claudin-like *hpo-30* (*Figure 2A–B*, *Figure 1—figure supplement 1A–D*), both of which are expressed and function in PVD during patterning of the dendritic arbor (*Liu and Shen, 2011*; *Smith et al., 2013*). In addition, we isolated mutant alleles in *tiam-1* (*Figure 2C*, *Figure 1—figure supplement 1E*), the *C. elegans* homolog (*Demarco et al., 2012*) of the multidomain vertebrate Rac1 guanine nucleotide exchange factor (GEF) Tiam1 (T-Lymphoma Invasion And Metastasis-Inducing Protein 1) (*Habets et al., 1994*). Transgenic expression of *tiam-1/GEF* with heterologous promoters in PVD neurons but not in other tissues efficiently rescued *tiam-1/GEF* mutant phenotypes, consistent with expression of a *tiam-1/GEF* reporter in PVD neurons (*Demarco et al., 2012*) (*Figure 1—figure supplement 1F–H*). Finally, we isolated a missense allele of *act-4/Actin* (*Figure 2D*, *Figure 1—figure supplement 1I*), one of five actins encoded in the *C. elegans* genome (*Krause et al., 1989*). This missense allele (*dz222*, G151E) is identical to the *wy917*

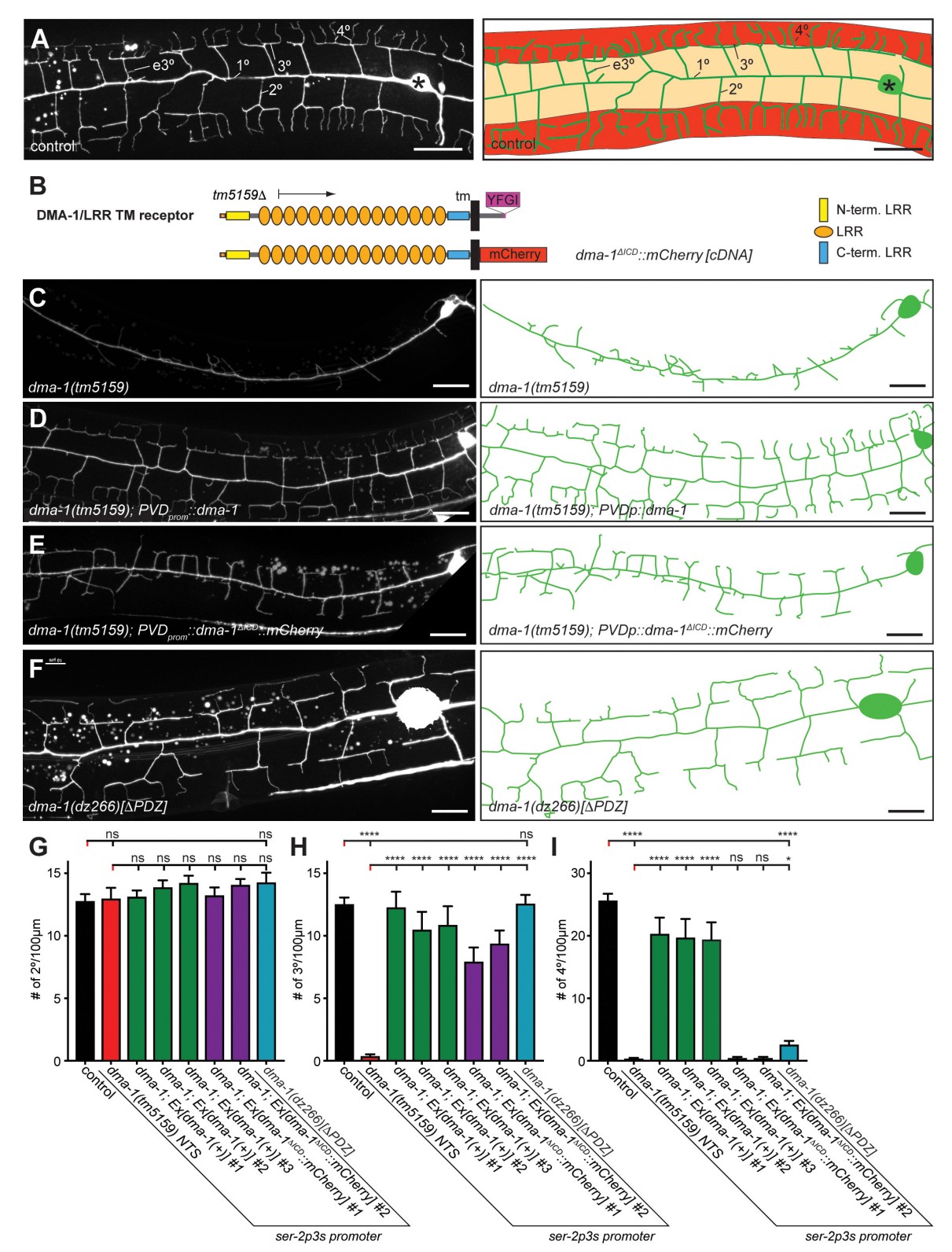

**Figure 1.** The intracellular domain of DMA-1/LRR-TM is required for higher order branching of PVD somatosensory dendrites. (A) Fluorescent images of PVD (left panels) and schematics (right panels) of wild-type control animals. PVD is visualized by the *wdIs52 [F49H12.4p::GFP]* transgene in all panels. 1°, 2°, 3°, 4°, and ectopic 3° (e3°) dendrites are indicated. Anterior is to the left and dorsal is up in all panels, scale bars indicate 20 μm. (B) Schematics of the DMA-1/LRR-TM protein and a variant used in transgenic rescue experiments (*dma-1(ΔICD)*). A PDZ-binding site (YFGI) at the extreme C-terminus of

*Figure 1 continued on next page*

*Figure 1 continued*

DMA-1/LRR-TM is indicated in lilac. The predicted deletion from the *tm5159* deletion allele is shown. (**C-F**) Fluorescent images of PVD (left panels) and schematics (right panels) of the genotypes indicated. Scale bar indicates 20 µm. (**G**) Quantification of 2°, 3°, and 4° branch numbers per 100 µm anterior to the PVD cell body. Data for three and two independent transgenic lines for the *dma-1* wild type cDNA or the *dma-1(ΔICD)*, respectively, are shown next to the data for the *dma-1(dz266[ΔPDZ])* allele. The data for *dma-1(tm5159) NTS* are nontransgenic siblings of a representative transgenic line. For raw data see *Figure 1—source data 1*. Data are represented as mean ± SEM. Statistical comparisons were performed using one-sided ANOVA with Sidak's correction. Statistical significance is indicated (ns, not significant; ****, p < 0.0001). n = 20 animals per genotype.

DOI: https://doi.org/10.7554/eLife.38949.002

The following source data and figure supplement are available for figure 1:

**Source data 1.** Complete source data.
DOI: https://doi.org/10.7554/eLife.38949.003
**Figure supplement 1.** Genes functioning cell-autonomously in PVD somatosensory neurons.
DOI: https://doi.org/10.7554/eLife.38949.004

allele identified in a related screen by the Shen lab (*Zou et al., 2018*). The G151E mutation is analogous to a dominant allele in *ACTA1* in a human patient with severe congenital myopathy where it resulted in abnormal actin aggregates (*Ravenscroft et al., 2011*). A reporter for *act-4* is expressed in muscle ((*Stone and Shaw, 1993*), data not shown), but fluorescence in situ hybridization experiments also suggested neuronal expression (*Birchall et al., 1995*). We found that transgenic expression of *act-4/Actin* in PVD but not in muscle rescued *act-4* mutant defects (*Figure 1—figure supplement 1J–M*). Additionally, expression of *act-1*, a paralog of *act-4*, rescued *act-4* mutant phenotypes (*Figure 1—figure supplement 1M*). All isolated mutant alleles also affected patterning of FLP neurons, a related pair of neurons, which cover the head region of the animal with similar dendritic arbors (*Figure 2—figure supplement 1A*). We conclude that in addition to DMA-1/LRR-TM and HPO-30/Claudin, TIAM-1/GEF and ACT-4/Actin function cell-autonomously to pattern the dendritic arbor of PVD and, likely FLP neurons. Moreover, expression of any actin in PVD rather than a specific function of ACT-4/Actin is important for dendrite patterning in PVD neurons.

## The PDZ-binding site of the DMA-1 leucine-rich transmembrane receptor (DMA-1/LRR-TM) is required for patterning of 4° branches but not 3° branches

Complete removal of *dma-1/LRR-TM* results in almost complete absence of 3° and 4° branches with additional effects on 2° branches (*Liu and Shen, 2011*; *Dong et al., 2013*; *Salzberg et al., 2013*) (*Figure 1B,C*, *Figure 2—figure supplement 2*). As previously shown (*Liu and Shen, 2011*), we found that the *dma-1* mutant phenotype can be fully rescued by transgenic expression of a wild type DMA-1/LRR-TM cDNA (*Figure 1C,D,G–I*). In contrast, a mutant where the intracellular domain (ICD) of DMA-1 was replaced by mCherry (ΔICD) resulted in partial rescue, where the number of 3° branches, but not 4° branches were restored (*Figures 1B,C,E,G–I*). The C terminus of DMA-1/LRR-TM follows the consensus sequence of a PDZ-binding site (X- Φ –X–Φ). We therefore used CRISPR/Cas9 genome editing to generate mutant animals predicted to express a form of DMA-1 lacking the last four residues (referred to as ΔPDZ thereafter). The number of 2° and 3° branches in *dma-1(ΔPDZ)* mutant animals appear indistinguishable from wild-type animals, whereas the number of 4° branches is significantly decreased (*Figure 1G–I*). These results indicate (1) that the intracellular domain of DMA-1/LRR-TM is required for the formation of PVD higher order branches, and (2) that the PDZ-binding site at the extreme C-terminus of DMA-1/LRR-TM is important for formation of 4° branches.

## The hpo-30/Claudin, tiam-1/GEF and act-4/Actin genes act in the menorin pathway

To genetically analyze the function of *dma-1/LRR-TM*, *hpo-30/Claudin*, *tiam-1/GEF* and *act-4/Actin*, we investigated the PVD mutant phenotype in single and double mutants using morphometric analyses. We determined the number and aggregate length for all classes of dendrites in a segment 100 µm anterior to the PVD cell body in different genetic backgrounds. We found that double mutants between *sax-7/L1CAM* and *hpo-30/Claudin* were not more severe than the more severe of the single mutants, indicating that *hpo-30/Claudin* functions in a pathway with *sax-7/L1CAM* for 2° branch

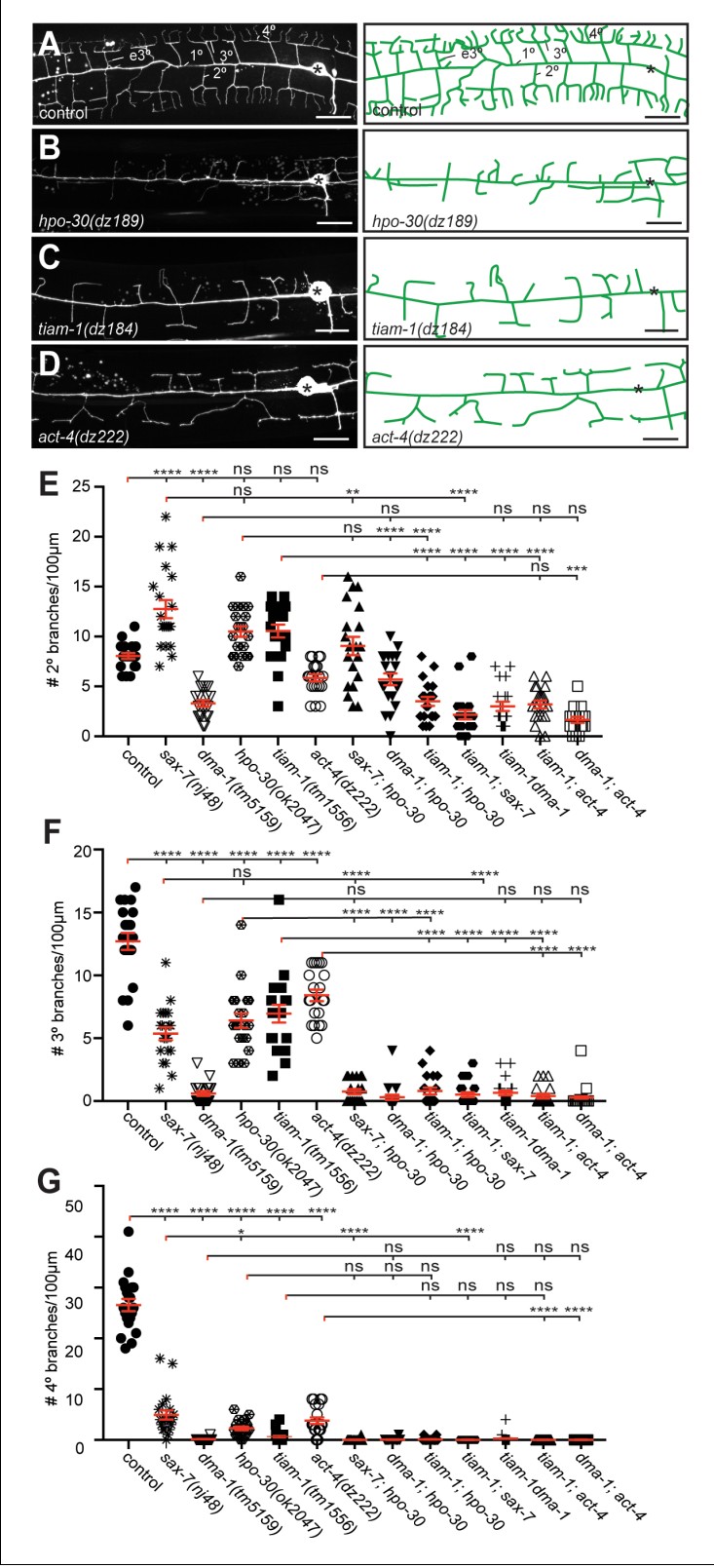

**Figure 2.** The *hpo-30/Claudin*, *tiam-1/GEF*, and *act-4/Actin* act genetically in the Menorin pathway. (**A – D**) Fluorescent images of PVD (left panels, visualized by the *wdIs52 [F49H12.4p::GFP]* transgene) and schematics (right panels) of the genotypes indicated. The control image is identical to *Figure 1A* and shown for comparison only. Details on alleles of individual genes and images of other alleles are shown in *Figure 1—figure supplement*
*Figure 2 continued on next page*

*Figure 2 continued*

*1*. Scale bar indicates 20 μm. (E – G) Quantification of 2°, 3°, and 4° branch numbers per 100 μm anterior to the PVD cell body. Data for additional single and double mutants of the Menorin pathway as well as the average length and aggregate length of secondary, tertiary, and quaternary branches are shown in *Figure 2—figure supplement 2*. All alleles used are molecular or genetic null alleles (Material and methods for details). Data for control, *lect-2*, *sax-7*, *mnr-1* and *dma-1* mutant animals is identical to data from *Díaz-Balzac et al. (2016)* and shown for comparison only. For raw data see *Figure 1—source data 1*. Data are represented as mean ± SEM. Statistical comparisons were performed using one-sided ANOVA with the Tukey correction. Statistical significance is indicated (ns, not significant; *, p<0.05; **, p<0.01; ***, p<0.001; ****, p<0.0001). n = 20 animals for all genotypes.
DOI: https://doi.org/10.7554/eLife.38949.005

The following figure supplements are available for figure 2:

**Figure supplement 1.** Effects of *hpo-30/Claudin*, *tiam-1/GEF* and *act-4/Actin* on localization of LECT-2:mNG, SAX-7::GFP and DMA-1::GFP.
DOI: https://doi.org/10.7554/eLife.38949.006

**Figure supplement 2.** The Genetics of *hpo-30/Claudin*, *tiam-1/GEF*, and *act-4/Actin*.
DOI: https://doi.org/10.7554/eLife.38949.007

patterning (*Figure 2E*). The *dma-1/LRR-TM; hpo-30/Claudin* and *dma-1/LRR-TM; tiam-1/GEF* were statistically indistinguishable from the *dma-1/LRR-TM* single mutant, suggesting that *dma-1/LRR-TM* is epistatic and required for most if not all functions during patterning of higher order branches in PVD. Interestingly, double mutants between *tiam-1/GEF* and *mnr-1/Menorin*, *lect-2/Chondromodulin II*, *kpc-1/Furin*, *sax-7/L1CAM*, *hpo-30/Claudin*, or *act-4*, respectively, appeared more severe than either of the single mutants alone, but indistinguishable from the *dma-1/LRR-TM* single mutant (*Figure 2E*, *Figure 2—figure supplement 2*). These findings suggest that *tiam-1/GEF* also serves in a genetic pathway that functions in parallel to *mnr-1/sax-7/lect-2/hpo-30* and, possibly, *act-4*. Similar genetic relationships were observed regarding the number of 3° branches and the aggregate length of 2° and 3° branches with one notable exception. Double mutants between *sax-7/L1CAM* and *hpo-30/Claudin* displayed a phenotype for 3° branches that was statistically more severe than either of the single mutants, yet indistinguishable from the *dma-1/LRR-TM* mutant (*Figure 2F*, *Figure 2—figure supplement 2*), suggesting a parallel function for HPO-30/Claudin for higher order branching. Common to all single and double mutants was the near complete absence of 4° branches (*Figure 2G*, *Figure 2—figure supplement 2*). Together, our results suggest that (1) *hpo-30/Claudin* and *tiam-1/GEF* act in the Menorin pathway to pattern PVD dendritic arbors and, that (2) *hpo-30/Claudin* and *tiam-1/GEF* may also serve independent functions.

We next sought to place *dma-1/LRR-TM*, *hpo-30/Claudin*, *tiam-1/GEF*, and *act-4/Actin* within the genetic pathway using a combination of gain and loss of function approaches. Previous work showed that expression of the hypodermally derived cell adhesion molecule *mnr-1/Menorin* in muscle of wild-type animals results in the appearance of dendritic arbors that resembled African Baobab trees (*Figure 3A*) (*Salzberg et al., 2013*). We used this *mnr-1/Menorin* gain of function (*gof*) phenotype in combination with loss-of-function mutations in *hpo-30/Claudin*, *tiam-1/GEF* and *act-4/Actin*. We found that loss of *hpo-30/Claudin*, *tiam-1/GEF* or *act-4/Actin* suppressed the formation of baobab-like dendritic arbors (*Figure 3A–E*) to the same extent as removal of other genes in the Menorin pathway such as *dma-1/LRR-TM*, *sax-7/L1CAM*, or *lect-2/Chondromodulin II* (*Salzberg et al., 2013*; *Díaz-Balzac et al., 2016*). Moreover, protein localization of a LECT-2/Chondromodulin II or SAX-7/L1CAM reporter is not visibly affected in *hpo-30/Claudin*, *tiam-1/GEF* or *act-4/Actin* mutants (*Figure 2—figure supplement 1B,C*). These findings suggest that *hpo-30/Claudin*, *tiam-1/GEF* and *act-4/Actin*, just like *dma-1/LRR-TM*, *sax-7/L1CAM*, or *lect-2/Chondromodulin II* function genetically downstream of or in parallel to *mnr-1/Menorin*.

We next asked in which order *dma-1/LRR-TM*, *tiam-1/GEF* and *act-4/Actin* function in PVD dendrites. Overexpression of *tiam-1/GEF* (*tiam-1(o/e)*) in an otherwise wild type background resulted in an increase of 2°, 3°, and 4° branches in PVD dendrites (*Figure 3F,G*). This excessive branching was completely suppressed by a mutation in *act-4/Actin*. Specifically, the *tiam-1(o/e); act-4* double mutant was statistically indistinguishable from the *act-4* single mutant, both with regard to the number and aggregate length of dendritic branches (*Figure 3H–J*, *Figure 3—figure supplement 1A–C*).

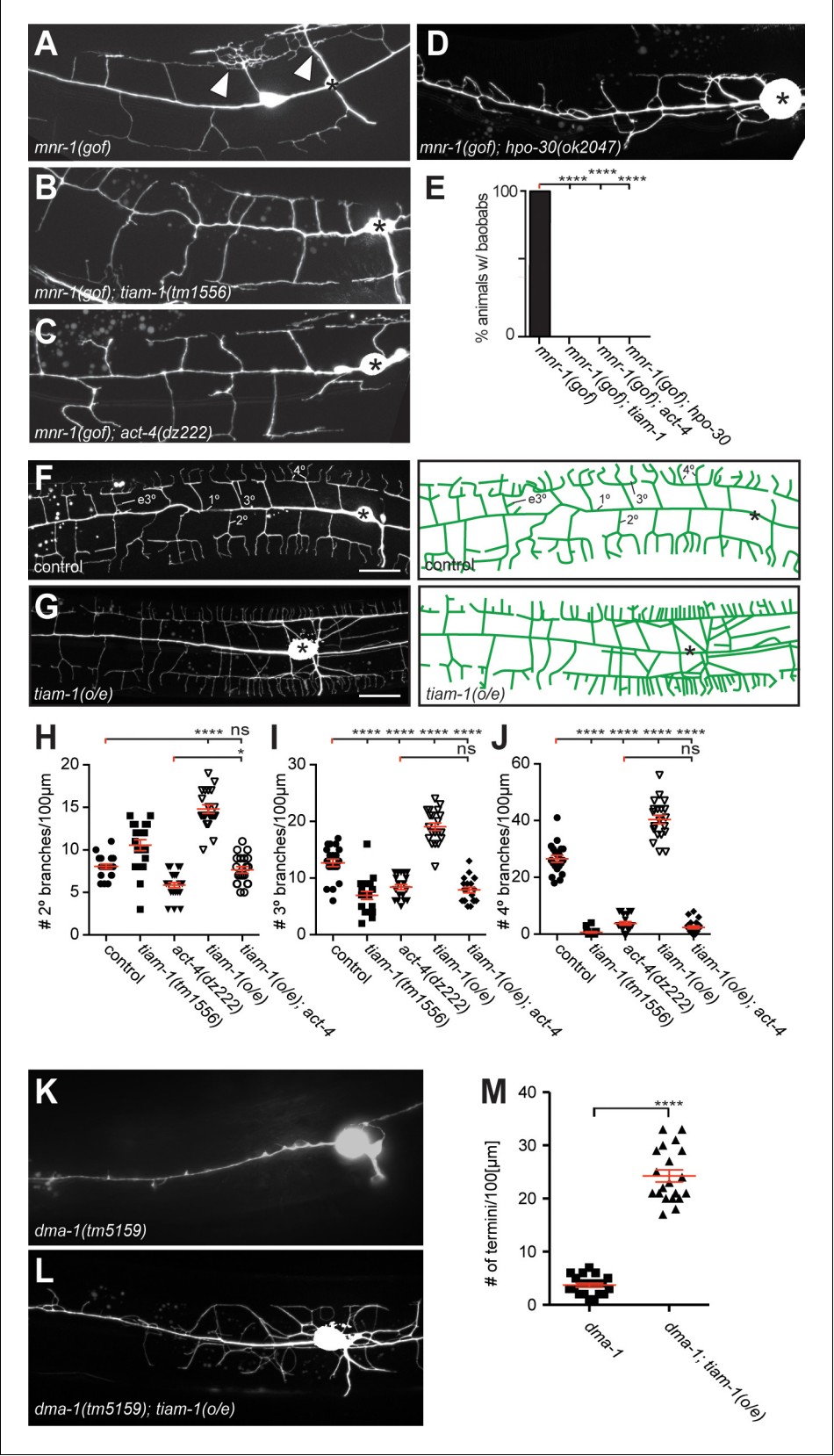

**Figure 3.** The *tiam-1/GEF* and *act-4/Actin* act downstream of the *dma-1/LRR-TM* receptor in PVD dendrites. (A – D) Fluorescent images of animals in which *mnr-1/Menorin* was expressed ectopically in muscle (*mnr-1(gof): dzIs43*, *Figure 3 continued on next page*

*Figure 3 continued*

[*myo-3p::mnr-1*] (*Salzberg et al., 2013*)) in different genetic backgrounds. A characteristic baobab-like tree is indicated by a white arrowhead in (**A**). PVD is visualized by the *wdIs52 [F49H12.4p::GFP]* transgene in all panels. Anterior is to the left, dorsal up, and scale bars indicate 20 µm in all panels. (**E**) Quantification of animals with baobab-like dendritic trees in the genotypes indicated. Data are represented as mean. For raw data see *Figure 1— source data 1*. Statistical comparisons were performed using the Z-test. Statistical significance is indicated (****p<0.0005). n = 20 animals for all samples. (**F – G**) Fluorescent images of PVD (left panels) and schematics (right panels) of wild type control and *tiam-1(o/e)* overexpressing animals (*dzIs95 [ser-2p3::tiam-1]*). The control image (**F**) is identical to *Figure 1A* and shown for comparison only. (**H – J**) Quantification of secondary (2°, (**H**), tertiary (3°, (**I**), and quaternary (4°, (**J**) branch numbers per 100 µm anterior to the PVD cell body in the genotypes indicated. Data for control, *tiam-1(tm1556)*, and *act-4(dz222)* are identical to data in *Figure 2* and shown for comparison only. For raw data see *Figure 1—source data 1*. Data are represented as mean ±SEM. Statistical comparisons were performed using one-sided ANOVA with the Tukey correction. Statistical significance is indicated (ns, not significant; *, p<0.05; ****p<0.0001). n = 20 for all samples. (**K – L**) Fluorescent images of PVD in *dma-1/LRR-TM* mutant animals alone (**K**) and in combination with a *tiam-1(o/e)* expression array (**L**). Anterior is to the left and ventral down. Scale bar indicates 20 µm. (**M**) Quantification of dendrite termini in a 100 µm section anterior to the PVD cell body in the genotype indicated. For raw data see *Figure 1—source data 1*. Data are represented as mean ±SEM. Statistical comparisons were performed using Student's T-test. Statistical significance is indicated (****p<0.0001). n = 20 animals for all samples.

DOI: https://doi.org/10.7554/eLife.38949.008

The following figure supplement is available for figure 3:

**Figure supplement 1.** The *tiam-1/GEF* and *act-4/Actin* act downstream or in parallel of the *dma-1/LRR-TM* receptor in PVD dendrites.

DOI: https://doi.org/10.7554/eLife.38949.009

These observations suggest that *tiam-1/GEF* function is dependent on ACT-4/Actin in PVD dendrites. We further hypothesized that *tiam-1/GEF* would function downstream of the *dma-1/LRR-TM* receptor. If that were the case, one would predict that loss of branching in *dma-1/LRR-TM* null mutants would be at least partially suppressed (i.e. reversed) by overexpression of *tiam-1/GEF.* Indeed, overexpression of *tiam-1(o/e)* in *dma-1/LRR-TM* null mutants significantly increased the number of branches in PVD dendrites (*Figure 3K–M*). Taken together, our experiments suggest a pathway, in which TIAM-1/GEF functions downstream of or in parallel to DMA-1/LRR-TM (and possibly *hpo-30/Claudin*) and in a manner that depends on ACT-4/Actin.

## HPO-30/Claudin regulates DMA-1/LRR-TM surface expression and trafficking

Both DMA-1/LRR-TM and HPO-30/Claudin are predicted transmembrane proteins functioning in PVD dendrites. To determine the mechanistic relationship between these two factors, we asked where functional DMA-1::GFP and a HPO-30::tagBFP reporter fusions are localized in PVD neurons. As previously reported (*Dong et al., 2016*), the DMA::GFP reporter is localized in a punctate fashion in the cell body and along the dendritic tree as well as on the membrane, visible as diffuse membrane staining throughout the whole dendritic tree (*Figure 4B*). We found that an HPO-30::tagBFP fusion also exhibited localized punctate and diffuse membrane staining along the dendritic processes with less staining in the axon of PVD neurons (*Figure 4A*), much like other recently described HPO-30::GFP reporters (*Smith et al., 2013*; *Zou et al., 2015*). The DMA-1::GFP and HPO-30:: tagBFP puncta show significant but not complete colocalization in dendritic processes, suggesting that at least some fraction of these proteins are closely associated (*Figure 4C–D*).

To determine if DMA-1/LRR-TM and HPO-30/Claudin are part of the same biochemical complex, we conducted co-immunoprecipitation experiments by transfecting human embryonic kidney cells (HEK293T) with DMA-1 (tagged with HA) and HPO-30 (tagged with V5) (*Figure 4E*) alone and in combination. We found that DMA-1.HA efficiently co-precipitated HPO-30.V5 from a cellular lysate, suggesting that both proteins are part of the same biochemical complex (*Figure 4F*). This biochemical interaction was independent of the intracellular domain of DMA-1 (*Figure 4G*), suggesting that the proteins interact through either the transmembrane segments or the extracellular domains of DMA-1.

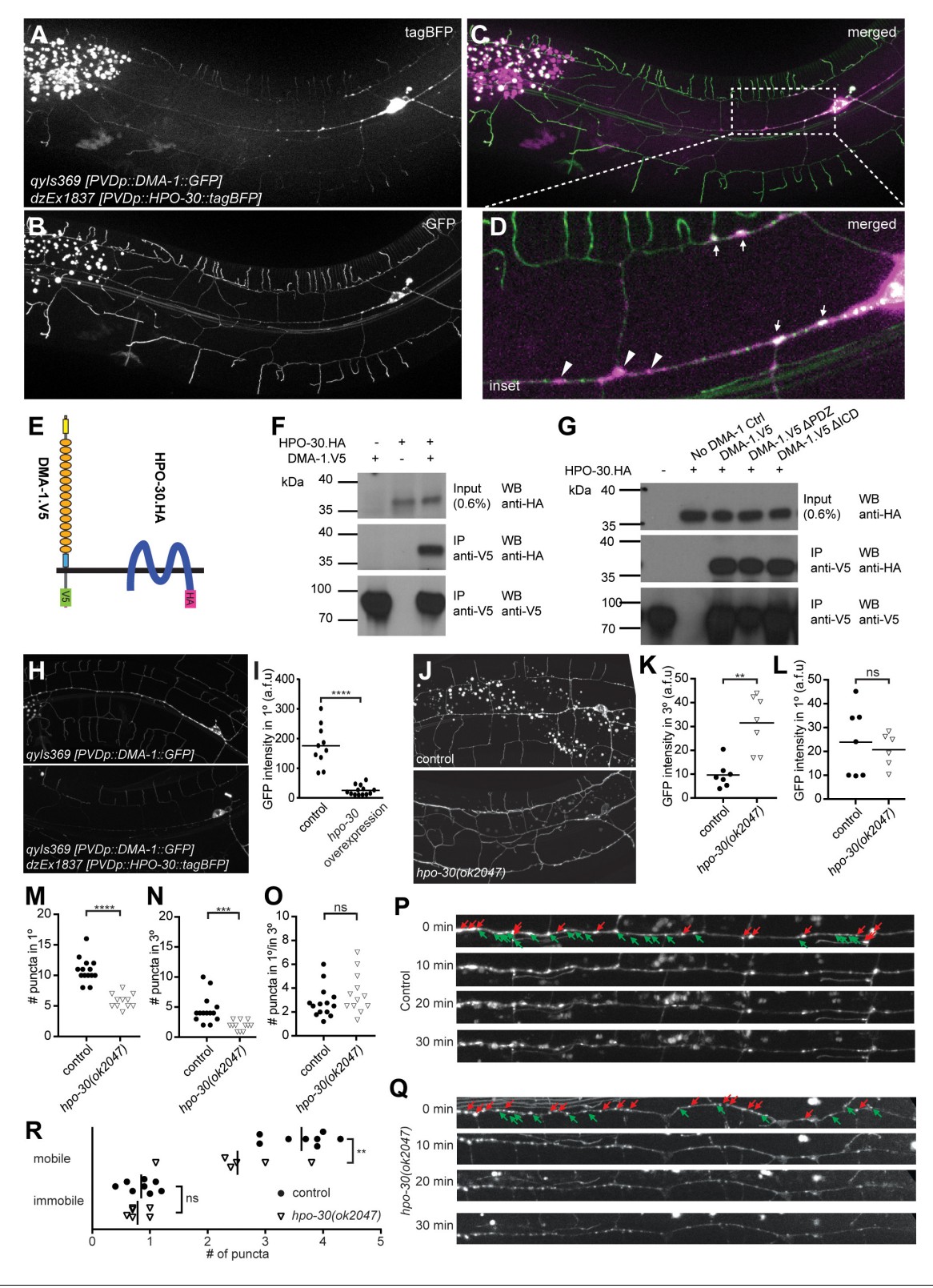

**Figure 4.** HPO-30/Claudin forms a complex with, and, localizes DMA-1/LRR-TM to higher order branches of PVD somatosensory neurons. (A – D) Images of an animal with PVD-specific expression of HPO-30::tagBFP (A, *dzEx1837*) and DMA-1::GFP (B, *qyIs369*) as well as a composite image (C), respectively. Both reporters show diffuse staining across the dendrite in addition to discreet punctate staining, with inset enlargement (D) where white arrows indicate puncta of HPO-30::tagBFP and DMA-1::GFP colocalization and a scale bar 10 μm. (E) Schematic showing the topography of the DMA-1/
*Figure 4 continued on next page*

*Figure 4 continued*

LRR-TM single pass transmembrane receptor and the four transmembrane, claudin-like, HPO-30 protein. Immuno tags (V5 and HA) used for co-immunoprecipitation experiments are shown. Not to scale. (F – G) Western blots of co-immunoprecipitation experiments. Transfected constructs are indicated above the panels. Antibodies used for immunoprecipitation (IP) and western blotting (WB) are indicated. A molecular marker is denoted on the left. Note, that all cases the two lower panels are from a single western blot, which was developed repeatedly with two different antibodies after stripping. In G., DMA-1.HA ΔPDZ and DMA-1.HA ΔICD are lacking the PDZ-binding site or the intracellular domain, respectively. Similar results were obtained from at least three independent replicate experiments. (H – I) PVD-specific expression of DMA-1::GFP (*qyIs369*) imaged in the presence (upper panel) or absence (lower panel) of a transgene with PVD-specific overexpression of HPO-30::tagBFP (*dzEx1837*) using identical imaging parameters for comparison of fluorescent intensity. The intensity of DMA-1::GFP appears substantially dimmer on the 1° dendrites when HPO-30::tagBFP is overexpressed. The average fluorescent intensity of the 1° dendrites within 50 µm of the cell body is quantified and presented in I). ****, $p < 0.0001$, Mann-Whitney test. (J – O). PVD-specific expression of DMA-1::GFP (*qyIs369*) imaged in a *hpo-30(ok2047)* null mutant or wild type control animal using identical imaging parameters for comparison of fluorescent intensity. Fluorescent intensity was quantified for 3° and 1° dendrites within 50 µm of the cell body in K and L, respectively. The number of puncta for 3° and 1° dendrites between 30 and 130 µm from the cell body was counted and presented in M and N, respectively; the ratio of the puncta between 3° and 1° dendrites is presented in O. For raw data see *Figure 1— source data 1*. Statistical significance is indicated as ns, not significant; **, $p < 0.01$; ***, $p < 0.001$; ****, $p < 0.0001$; Mann-Whitney test. (P) – R. Images of selected time points in a time-lapse series of visualizing DMA-1::GFP (*qyIs369*) in wild-type control and *hpo-30* null mutant animals, imaged in 2 min intervals for 30 min to observe the dynamics of DMA-1::GFP puncta. Puncta between 30 to 130 µm anterior to the cell body were identified at t = 0 min and classified as mobile or immobile. Puncta were deemed immobile if there was no perceived movement throughout the whole time-lapse assay (30 min), and mobile otherwise. The number of mobile and immobile puncta were quantified in R. For raw data see *Figure 1— source data 1*. Statistical significance is indicated as **, $p < 0.01$, 2-way ANOVA.

DOI: https://doi.org/10.7554/eLife.38949.010

We noticed that overexpression of the HPO-30::tagBFP reporter in PVD resulted in significantly reduced fluorescent intensity of the DMA-1::GFP reporter in 1° dendrites compared to control animals (*Figure 4H,I*). Conversely, loss of *hpo-30/Claudin* resulted in an increase of diffuse dendrite staining of the DMA-1::GFP reporter in 3° but not in 1° dendrites (Figure J-L). In addition, we found the number of DMA-1::GFP puncta in 1° and 3° branches reduced in *hpo-30* null mutants, although their relative distribution remained unchanged (*Figure 4M–O*). It has been suggested that the diffuse staining of DMA-1::GFP represents the surface membrane fraction while punctate staining represents the vesicular fraction (*Taylor et al., 2015*; *Zou et al., 2015*). Therefore, our results suggest that the normal function of HPO-30/Claudin is to negatively regulate the DMA-1/LRR-TM surface membrane fraction and to increase the vesicular fraction.

To further investigate this notion, we carried out time lapse imaging of the DMA-1::GFP reporter for 30 min in both wild type and *hpo-30* mutant animals (*Figure 4P–Q*). We found that in both genotypes, DMA-1::GFP puncta fell into a mobile and a stationary fraction, respectively. The mobile fraction exhibited a wide range of moving speeds in both retrograde and anterograde directions. Intriguingly, the number of mobile DMA-1::GFP puncta was decreased in *hpo-30* mutant animals compared to wild type animals, while the number of immobile DMA-1::GFP puncta remained unchanged (*Figure 4R*). The mobile DMA-1::GFP puncta may be vesicles that transport synthesized or internalized DMA-1::GFP, while immobile puncta may be vesicles or membrane fraction aggregates that serve signaling purposes. Collectively, we conclude that (1) HPO-30/Claudin exists in a complex with DMA-1/LRR-TM, that (2) HPO-30 negatively regulates DMA-1::GFP membrane levels and, that (3) HPO-30 serves a function in trafficking of DMA-1::GFP vesicles.

## The TIAM-1/GEF guanine nucleotide exchange factor appears to function independently of its Rac1 GEF activity to pattern PVD dendrites

To investigate the functions downstream of DMA-1/HPO-30, we focused next on the function of the guanine nucleotide exchange factor TIAM-1/GEF. *C. elegans* TIAM-1/GEF is a multidomain protein (*Figure 5A*), which comprises, in order, a myristoylation signal, an EVH1 (Ena/Vasp homology) domain, a PDZ domain, and a DH/PH (Dbl homology domain/plekstrin homology domain) encoding the enzymatic guanine nucleotide exchange activity (*Demarco et al., 2012*). Consistent with work in flies and vertebrate neurons (*Sone et al., 1997*; *Kunda et al., 2001*; *Tolias et al., 2005*), *C. elegans* TIAM-1/GEF can shape neurons through mechanisms dependent on the small GTPases *mig-2/RhoG* and *ced-10/Rac1* (*Demarco et al., 2012*). In addition to *mig-2/RhoG* and *ced-10/Rac1*, the *C. elegans* genome encodes one additional Rac1-like GTPase, *rac-2* (*Lundquist et al., 2001*). Surprisingly,

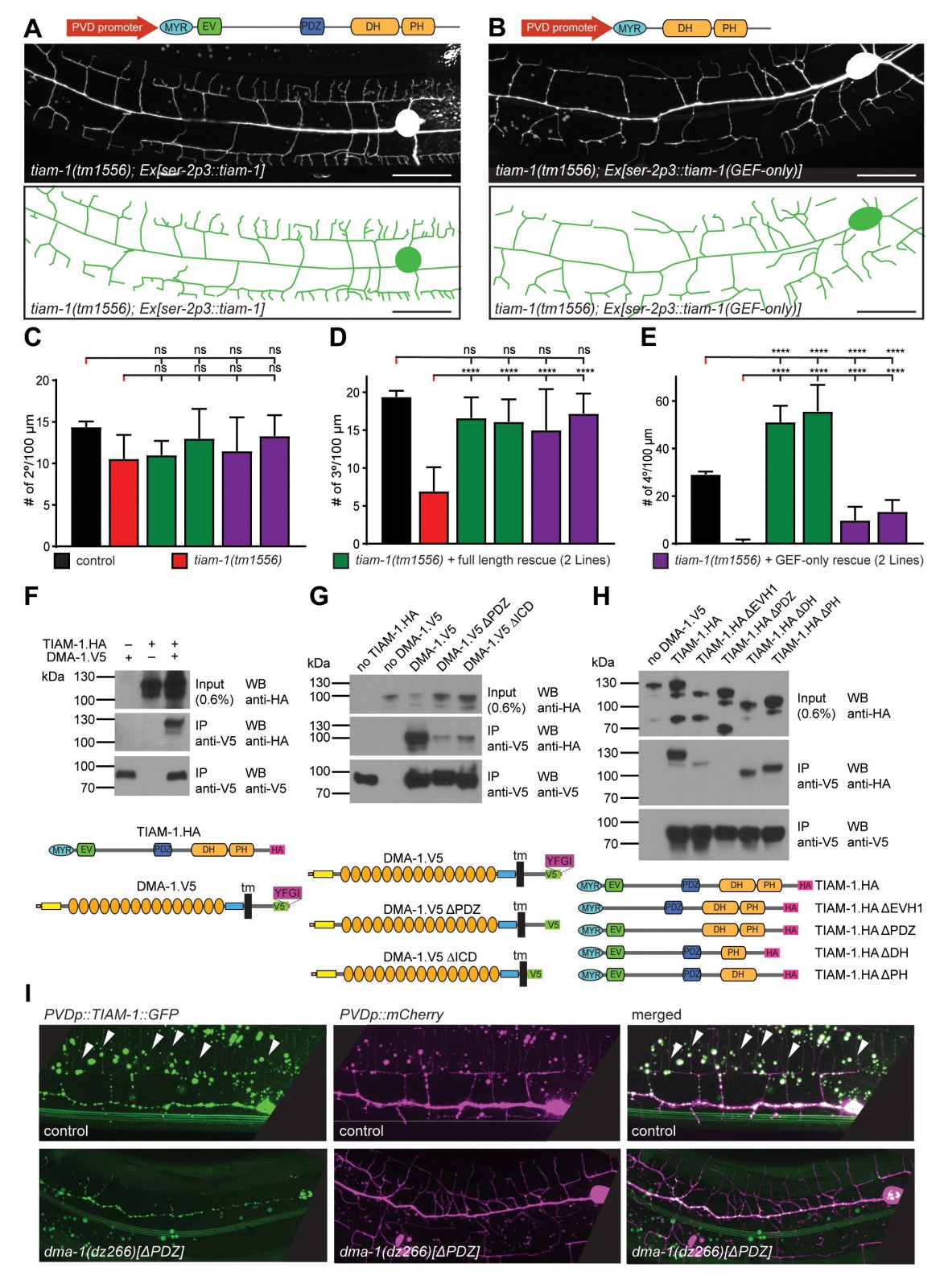

**Figure 5.** TIAM-1/GEF interacts with DMA-1/LRR-TM through PDZ motif/domain interaction to shape 4° dendrite branches. (**A – B**) Fluorescent image with schematics of *tiam-1* mutant animals carrying either a full length or deletion construct (GEF-only) of the *tiam-1* cDNA under control of the PVD-specific *ser-2p3* promoter. (**B**) (**C – E**) Quantification of 2°, 3°, and 4° branch numbers per 100 μm anterior to the PVD cell body in the genotypes indicated. For raw data see *Figure 1 — source data 1*. Statistical significance is indicated as Data are represented as mean ±SEM. Statistical

*Figure 5 continued on next page*

*Figure 5 continued*

comparisons were performed using one-sided ANOVA with the Tukey correction. Statistical significance is indicated (****p<0.0005). n = 10 animals per genotype. (F – H) Western Blots of co-immunoprecipitation experiments with corresponding schematic showing the topography of the DMA-1/LRR-TM single pass transmembrane receptor and the TIAM-1/GEF multidomain protein. Schematics not to scale. Immuno tags (V5 and HA) used for co-immunoprecipitation experiments are shown. Transfected constructs are indicated above the panels. Antibodies used for immunoprecipitation (IP) and western blotting (WB) are indicated. A molecular marker is on the left. Note, that the two lower panels are from a single western blot, which was developed repeatedly with two different antibodies after stripping in all panels. Panel F investigates the interaction between TIAM-1 and DMA-1, and panels G and H are structure function analyses to delineate the domains required for the TIAM-1/DMA-1 interaction in TIAM-1 and DMA-1, respectively. Similar results were obtained from at least three independent replicate experiments. (G) (H) (I) Fluorescent images of TIAM-1::GFP reporter expressed in PVD in wild type or *dma-1(Δ|DZ)* mutant animals. PVD is visualized by *dzIs53 ([F49H12.4p::mCherry])*. TIAM-1::GFP (*wyIs1139*) staining on 4° dendrite branches in wild-type animals is lost in *dma-1(ΔPDZ)* mutant animals.
DOI: https://doi.org/10.7554/eLife.38949.011

The following figure supplements are available for figure 5:

**Figure supplement 1.** The TIAM-1/GEF acts independently of Rac1 GEF activity.
DOI: https://doi.org/10.7554/eLife.38949.012
**Figure supplement 2.** Supplementary co-immunoprecipitation experiments (related to *Figure 5* (A) *Figure 7* (B)).
DOI: https://doi.org/10.7554/eLife.38949.013

---

neither mutations in *mig-2/RhoG* and *ced-10/Rac1* alone or in combination, nor a mutation in *rac-2* resulted in obvious phenotypes in PVD dendritic arbor formation (*Figure 5—figure supplement 1A–C*), implying that TIAM-1/GEF could function independently of these Rac1-like GTPases, at least individually. To further investigate this notion, we conducted transgenic rescue experiments with deletions and point mutant variants of TIAM-1/GEF. We found that full length TIAM-1/GEF alone or as a C-terminal fusion with mCherry fully rescued the PVD dendrite patterning defects (*Figure 5—figure supplement 1D*). Surprisingly, a T548F point mutant of full length TIAM-1/GEF still rescued the PVD mutant phenotype (*Figure 6B*, *Figure 5—figure supplement 1D*). The fact that the very same mutation in the *C. elegans* TIAM-1 DH/PH domain (*Demarco et al., 2012*) as well as the analogous point mutation in the UNC-73/Trio GEF abolishes Rac1 GEF activity in vitro (*Steven et al.,*

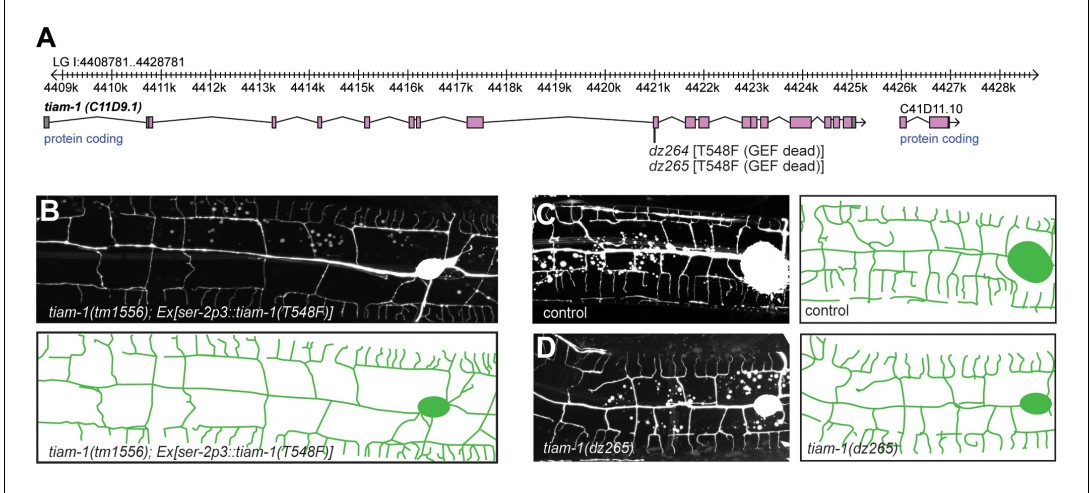

**Figure 6.** The TIAM-1/GEF functions independently of its guanine nucleotide activity. (**A**) Genomic environs of the *tiam-1* locus in linkage group I (LGI) with exons and introns indicated. The location of the two CRISPR/Cas9 engineered *dz264* and *dz265* alleles, encoding a missense mutation that causes the T548F mutation is shown. (**B**) Fluorescent images with schematics of *tiam-1* mutant animals carrying a T548F point mutant form of the *tiam-1* cDNA under control of the PVD-specific *ser-2p3* promoter. The T548F point mutant is analogous to the S1216F mutation in UNC-73 (*Steven et al., 1998*) and the T516F mutation in a previously described TIAM-1 cDNA (*Demarco et al., 2012*). Importantly, the point mutations in both proteins have been shown to abrogate GEF activity toward Rac in vitro (*Demarco et al., 2012*). (**C – D**) Fluorescent images of PVD (left panels) and schematics (right panels) of wild-type control (**C**) and *tiam-1(dz265)* mutant animals (**D**). PVD is visualized by the *wdIs52 [F49H12.4p::GFP]* transgene in all panels.
DOI: https://doi.org/10.7554/eLife.38949.014

*1998*), lent further support to the idea that TIAM-1/GEF activity may not be required for PVD patterning (*Figure 6B*). It was conceivable, however, that mutant TIAM-1(T548F) retained some residual GEF activity that through high level transgenic expression could provide sufficient enzymatic GEF activity to rescue the PVD dendrite defects of *tiam-1* mutants. To exclude this possibility, we used CRISPR/Cas9 genome editing to engineer the T548F missense mutation into the endogenous *tiam-1/GEF* locus. Two independently isolated alleles (*dz264* and *dz265*) that harbored the T548F point mutation in the *tiam-1/GEF* locus appeared phenotypically normal in regard to PVD patterning (*Figure 6C–D* and data not shown). Thus, a point mutant form of TIAM-1 (T548F) devoid of measurable Rac1 GEF activity in vitro is sufficient for PVD patterning in vivo. In summary, our results suggest that TIAM-1/GEF can function independently of its GEF enzymatic activity although we can not formally exclude that the TIAM-1(T548F) shows enzymatic activity towards small GTPases other than Rac1 that may be involved in PVD patterning.

## The TIAM-1/GEF guanine nucleotide exchange factor interacts with DMA-1/LRR-TM through a PDZ ligand motif/domain to promote 4° branching

Since the PDZ ligand motif of DMA-1/LLR-TM is important for the formation of 4° branches (*Figure 1*), we tested if the PDZ domain of TIAM-1/GEF may mediate this function. We found that truncated TIAM-1, lacking the EV and PDZ domains (referred to as GEF-only), only partially rescued the PVD mutant phenotype (*Figure 5B*, *Figure 5—figure supplement 1*). Specifically, the GEF-only construct rescued formation of PVD 2° and 3° defects in the same manner as the full-length TIAM-1 construct, but failed to rescue defects in the number of 4° branches (*Figure 5C–E*). Intriguingly, overexpression of full length TIAM-1 resulted in more 4° branches, suggesting that TIAM-1 function is dosage dependent (*Figure 5E*). Thus, the DH/PH domain of TIAM-1 is sufficient for 2° and 3° branch formation, but not for 4° branch formation.

The remarkable similarity of phenotypes, namely a reduction of 4° branches, in the DMA-1(ΔPDZ) allele (*Figure 1*) and TIAM-1(GEF-only) rescued animals (*Figure 5*) suggested that DMA-1/LRR-TM and TIAM-1/GEF may function through a PDZ interaction for the formation of 4° branches. To test this hypothesis, we transfected DMA-1 (tagged with V5, DMA-1.V5) and TIAM-1 (tagged with HA, TIAM-1.HA) in human embryonic kidney cells (HEK293T) and conducted co-immunoprecipitation experiments from lysates of transiently transfected cells. We found that DMA-1.V5 efficiently co-immunoprecipitated TIAM-1.HA from lysates (*Figure 5F*). This interaction was dependent on the DMA-1 intracellular domain (ICD) and, specifically on the PDZ binding site, as the interaction was strongly reduced if either was removed (*Figure 5G*). Moreover, the interaction between TIAM-1/GEF and DMA-1/LRR-TM appeared specific, because replacing the DMA-1 PDZ binding site (YFGI) with the heterologous PDZ motif (EFYA) of the heparan sulfate proteoglycan SDN-1/Syndecan reduced the interaction between TIAM-1 and DMA-1 to the same extent as removing the PDZ binding site entirely (*Figure 5—figure supplement 2A*). Residual binding activity may be contributed by the EVH1 domain, as removing the EVH1 domain from TIAM-1/GEF also compromised the interaction with DMA-1 (*Figure 6H*). On the other hand, removing the PDZ domain from TIAM-1 (ΔPDZ) completely abrogated the interaction between DMA-1/LRR-TM and TIAM-1/GEF (*Figure 6H*). Collectively, our result showed (1) that TIAM-1/GEF interacts with DMA-1/LRR-TM through a PDZ domain-motif interaction with a possible contribution of the EVH1 domain in TIAM-1, and (2) that loss of PDZ domain or motif from either protein functionally result in a specific loss of 4° dendritic branches, suggesting that DMA-1/LRR-TM localizes TIAM-1/GEF to direct 4° dendrite growth. Consistent with this hypothesis, the localization of a TIAM-1::GFP reporter to 4° dendrite branches (but not lower branches) was lost in the *dma-1* mutant in which the PDZ motif of DMA-1 was deleted (*dma-1(dz266)[ΔPDZ]*) (*Figure 5I*).

## F-Actin is localized to the leading edge of dendrites

The filamentous F-actin and microtubule polymers are part of the cytoskeleton that provide stability and force during growth and development of neuronal processes (*Dent et al., 2011*; *Kapitein and Hoogenraad, 2015*). Having identified a function for *act-4/Actin* in PVD dendrite patterning, we sought to visualize F-actin in PVD neurons of live animals. To this end, we used the calponin homology domain of utrophin (UtrCH) fused to tagRFP (tagRFP::UtrCH), which has been previously shown

to faithfully visualize F-actin without effects on actin dynamics (*Burkel et al., 2007*; *Chia et al., 2014*). Transgenic animals expressing tagRFP::UtrCH in PVD neurons showed the reporter localized to defined subcellular compartments during development of the dendritic arbor. During the L2 larval stage tagRFP::UtrCH was primarily localized to the cell body, the 1°, 2°, and budding 3° branches (*Figure 7A*). During the subsequent larval stages, tagRFP::UtrCH remained strongly expressed in the cell body, but otherwise became successively more localized to distal (i.e. developing) branches, rather than proximal branches. For example, at the young adult stage, tagRFP::UtrCH was primarily localized to 4° branches, and less to 3° and 2° branches (*Figure 7A*). These results indicate that tagRFP::UtrCH, and by inference F-actin, localizes to extending branches. Conversely, an α-tubulin fusion (tagRFP::TBA-1) fusion that labels microtubules, specifically expressed in PVD neurons is primarily localized to axons and 1° dendrites of PVD neurons (*Figure 7—figure supplement 1*), as previously described (*Maniar et al., 2011*). Thus, microtubules and F-actin occupy non-overlapping compartments in dendrites. Interestingly, this largely mutually exclusive localization of F-actin and microtubules is lost in mutants of the Menorin pathway. For example, mutations in *dma-1/LRR-TM*, *hpo-30/Claudin*, *sax-7/L1CAM*, or *tiam-1/GEF* result in a relocalization of the UtrCH reporter, and by inference F-actin, to more proximal dendrites (*Figure 7B*) with more dendritic endings lacking F-actin in *tiam-1/GEF* mutants (*Figure 7C*). Conversely, the tagRFP::TBA-1 fusion can be found in more distal dendrites in mutants of the Menorin pathway (*Figure 7—figure supplement 1*). These results suggest that the Menorin pathway establishes polarity and may function to localize F-actin to growing dendrites.

## TIAM-1/GEF may link DMA-1/LRR-TM to ACT-4/Actin

Based on our genetic findings, we next investigated the biochemical relationship of DMA-1/LRR-TM and TIAM-1/GEF with ACT-4/Actin to form a biochemical complex. We found that TIAM-1 co-immunoprecipitated ACT-4 from lysates of cells transiently transfected with tagged constructs of TIAM-1 and ACT-4 (*Figure 7D*). Intriguingly, we found that the ACT-4 G151E missense mutation encoded by the *dz222* missense allele from our genetic screen, dramatically increased the strength of the biochemical interaction (*Figure 7E*). Deletion of individual domains of TIAM-1/GEF failed to abrogate the interaction with either wild type or G151E mutant ACT-4/Actin (*Figure 5—figure supplement 2B*), suggesting that more than one domain in TIAM-1/GEF may act in a partially redundant fashion to bind ACT-4/Actin. Alternatively, a multiprotein complex could account for this observation. To test this notion directly, we added DMA-1.V5 to the transfection mix of TIAM-1.HA and ACT-4. FLAG. Surprisingly, we found that ACT-4.FLAG could also be precipitated with an antibody against DMA-1.V5, with the interaction possibly stronger in the presence of TIAM-1/GEF (*Figure 7F*). Collectively, we conclude that TIAM-1/GEF is part of a complex containing both the DMA-1/LRR-TM receptor and ACT-4/Actin, potentially directly linking the receptor to the cytoskeleton.

## Discussion

### HPO-30/Claudin regulates DMA-1/LRR-TM surface expression and trafficking

We found that both membrane surface levels as well as trafficking of a DMA-1/LRR-TM reporter were affected by the amount of HPO-30/Claudin expressed in PVD. Loss of *hpo-30/Claudin* increased, whereas overexpression of *hpo-30/Claudin* decreased membrane surface staining of the DMA-1/LRR-TM reporter, respectively. In addition, loss of *hpo-30/Claudin* also reduced the fraction of mobile vesicles containing the DMA-1::GFP reporter within the dendrite. Together, these findings suggest that HPO-30/Claudin promotes internalization and trafficking of the cell surface receptor DMA-1/LRR-TM. This conclusion is also consistent with a recent study demonstrating that HPO-30/Claudin negatively regulates the level of the synaptic cell adhesion protein NLG-1/Neuroligin at the cholinergic NMJ (*Sharma et al., 2018*). There are several, not mutually exclusive, explanations for the functional significance of DMA-1/LRR-TM regulation by HPO-30/Claudin. First, the surface expression of DMA-1/LRR-TM may have to be tightly regulated, because overexpression of surface DMA-1/LRR-TM could result in too strong or sustained binding to its cognate binding partners, the hypodermally expressed cell adhesion molecules SAX-7/L1CAM and MNR-1/Menorin. Secondly, recycling of DMA-1 may be required for DMA-1 to remain functional. For example, cell adhesion

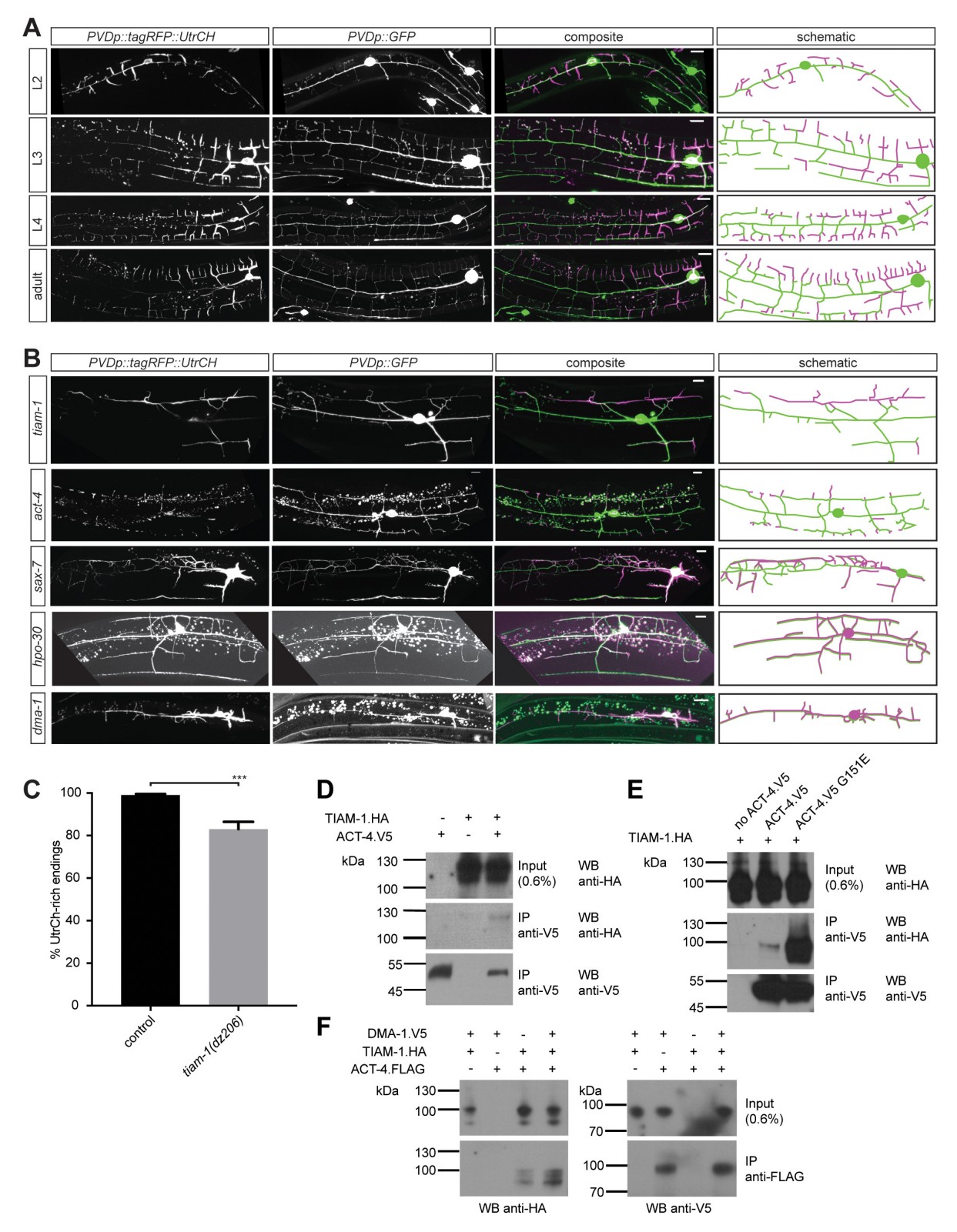

**Figure 7.** F-actin is localized to the leading edges of developing dendrites and requires the Menorin pathway for polarized localization. (A) Fluorescent images of animals at different developmental stages carrying a F-actin reporter (*ser-2p3::tagRFP::UtrCH [dzEx1564]*, left panels), a PVD cytoplasmic GFP reporter (*wdIs52 [F49H12.4p::GFP]*, second panels), merged images (third panel), and corresponding schematics (right panels). L2: second larval stage, L3: third larval stage, L4 fourth larval stage. Scale bars indicate 10 µm. (B) Fluorescent images of animals in different genetic backgrounds carrying a

*Figure 7 continued on next page*

*Figure 7 continued*

F-actin reporter (*ser-2p3::tagRFP::UtrCH [dzEx1564]*, left panels), a PVD cytoplasmic GFP reporter (*wdIs52 [F49H12.4p::GFP]*, second panels), merged images (third panel), and corresponding schematics (right panels). Genotypes are indicated on the left. Scale bars indicate 10 µm. (**C**) Quantification of dendrite termini with F-actin rich staining in wild-type control (n = 10) and *tiam-1(tm1556)* mutant animals (n = 13). For details on quantification see Materials and methods. For raw data see *Figure 1— source data 1*. Statistical significance is indicated as Data are represented as mean ±SEM. Statistical comparisons were performed using the Mann-Whitney test. Statistical significance is indicated (***, p<0.005). (**D – F**) Western blots of co-immunoprecipitation experiments analyzing the interaction between TIAM-1 and wild type ACT-4 or mutant ACT-4(G151E). Transfected constructs are indicated above the panels. Antibodies used for immunoprecptation (IP) and western blotting (WB) are indicated. A molecular marker is on the left. Note, that panels in F are from a single western blot, which was developed repeatedly with two different antibodies after stripping. Similar results were obtained from at least three independent replicate experiments.
DOI: https://doi.org/10.7554/eLife.38949.015

The following figure supplement is available for figure 7:

**Figure supplement 1.** Localization of TBA-1::tagRFP in different genetic backgrounds.
DOI: https://doi.org/10.7554/eLife.38949.016

molecules, such as cadherins are constantly recycled and, blocking endocytosis results in dysfunctional cell adhesion phenotypes (*Cadwell et al., 2016*). Alternatively, recycling of DMA-1/LRR-TM may be necessary for transporting DMA-1/LRR-TM to locations of particular demand, such as the tip of growing dendrites. Thirdly, the internalization of DMA-1/LRR-TM maybe part of the signal transduction itself, as internalization of surface receptors can be required to trigger signal transduction pathways occurring at a distance from the cell surface (*Sorkin and von Zastrow, 2009*).

## DMA-1/LRR-TM regulates 3° and 4° branch formation through distinct mechanisms

Several lines of evidence suggest that signaling downstream of the DMA-1/LRR-TM receptor diverges into at least two molecularly distinct, partially redundant pathways to establish 2° and 3° branches and 4° branches, respectively. First, removing the PDZ ligand motif of DMA-1/LRR-TM results in a decrease in 4° branches but not 2° or 3° branches. Conversely, a GEF-only construct of TIAM-1/GEF, which lacks the PDZ domain, rescues the 3° branching defects, but not 4° branching defects of *tiam-1* mutant animals. Thus, compromising the PDZ-mediated interaction between TIAM-1/GEF and DMA-1/LRR-TM results in specific loss of 4° branching. Furthermore, immunoprecipitation assays strongly indicate that TIAM-1/GEF and DMA-1/LRR-TM can interact directly through the PDZ motif at the C-terminus of DMA-1 and the PDZ-binding domain of TIAM-1. Since *tiam-1* overexpression resulted in excessive branching, TIAM-1/GEF may promote branching in a dose-dependent manner. We propose that the PDZ interaction with DMA-1 recruits TIAM-1/GEF to specific subcellular locations, where a locally higher concentration of TIAM-1/GEF induces dendrite branching. Indeed, this is supported by the observation that TIAM-1::GFP fails to localize to 4° branches in DMA-1(ΔPDZ) animals. These findings are also consistent with a recent study, which also suggested that a TIAM-1::GFP reporter expressed in PVD is recruited to higher order dendritic branches by DMA-1/LRR-TM (*Zou et al., 2018*). Collectively, these findings suggest that the PDZ-mediated interaction of TIAM-1/GEF and DMA-1/LRR-TM interaction is required for normal 4° dendrite formation.

An intriguing finding is that transgenic expression of a TIAM-1(GEF-only) construct lacking the PDZ interaction domain can rescue both the formation of 2° and 3° PVD branches. Conversely, DMA-1/LRR-TM lacking the cytoplasmic domain was still capable of rescuing 2° and 3° branching number (but not extension), suggesting that protein interactions of the DMA-1 transmembrane or the extracellular domain, possibly in conjunction with TIAM-1/GEF may mediate an alternative parallel pathway. Indeed, imaging experiments and biochemical assays suggest that (1) DMA-1/LRR-TM colocalizes with the HPO-30/Claudin and that (2) both proteins can form a biochemical complex through an interaction independent of the cytoplasmic domain of DMA-1. Moreover, genetic double mutant analyses show that *hpo-30* double mutants with *tiam-1* or *sax-7/L1CAM* display enhanced phenotypes consistent with functions in parallel pathways. Previous studies have shown that

mutating components of the WASP family verprolin-homologous protein (WVE-1/WAVE) regulatory complex (WRC) of actin regulators result in defects in PVD patterning (*Aguirre-Chen et al., 2011*; *Zou et al., 2018*). Moreover, HPO-30 can physically interact with the WRC and, compromising the molecular interactions between both DMA-1 and TIAM-1 and, between HPO-30 and the WRC at the same time result in synergistic defects consistent with the notion of parallel pathways downstream of the DMA-1 receptor (*Zou et al., 2018*). Zou *et al.* proposed that the DMA-1/LRR-TM receptor, in conjunction with HPO-30/Claudin and TIAM-1/GEF recruit actin regulators to the membrane to form dendritic branches. Alternatively, and consistent with our observation that HPO-30 negatively regulates the surface localization of the DMA-1/LRR-TM reporter, recruitment of the WRC by HPO-30 may also serve a role in endocytosis of DMA-1::GFP, as the WRC is known to play a role in clathrin-mediated endocytosis (*Mooren et al., 2012*). In possible support of such a function, the *act-4 (dz222)* allele results in more distal localization of DMA-1::GFP, including at the tip of growing dendrites (*Figure 2—figure supplement 1D,E*).

## The DMA-1/LRR-TM may shape 4° dendrites by directly regulating F-actin

F-actin clearly plays an important role in the formation of PVD dendritic branches. First, depolymerizing F-actin in vivo results in severe defects in PVD patterning (*Harterink et al., 2017*). Second, F-actin is localized to the distal tips of developing PVD dendrites and appears mutually exclusive with microtubules. Third, we find that F-actin localization to distal dendrites depends on the presence of transmembrane receptors on PVD dendrites and their ligands. Lastly, RNAi-mediated gene knock down of other genes with possible or established roles as regulators of F-actin in developing axons such as *unc-44/Ankyrin* (*Otsuka et al., 1995*), *unc-115/Ablim* (*Lundquist et al., 1998*; *Struckhoff and Lundquist, 2003*), or *unc-70/β-spectrin* (*Hammarlund et al., 2000*), display similar defects in PVD dendrite patterning as *act-4/Actin* and *tiam-1/GEF* mutants (*Aguirre-Chen et al., 2011*).

Our experiments show that the DMA-1/LRR-TM receptor is part of a biochemical complex with the TIAM-1/GEF and ACT-4/Actin. How could this complex mediate 4° dendrite growth or patterning? One possibility is that the DMA-1/TIAM-1/ACT-4 complex is part of a molecular clutch mechanism, first proposed as a means to explain actin-dependent cellular movements by *Mitchison and Kirschner (1988)*. These authors suggested that the force generated by polymerizing F-actin is transmitted through engagement of F-actin filaments with the membrane through transmembrane receptors. For example, transmembrane receptors of the Ig domain superfamily have been shown to couple extracellular interactions to cytoskeletal actin dynamics (reviewed in *Giannone et al. (2009)*). In analogy, binding of the DMA-1/LRR-TM receptor by the extracellular SAX-7/MNR-1/LECT-2 ligand complex could result in recruitment of TIAM-1/GEF, ACT-4/Actin and DMA-1/LRR-TM in a complex, thereby directly linking the receptor to the cytoskeleton. The interaction between TIAM-1/GEF and ACT-4/Actin is likely specific, because animals carrying the *act-4(dz222)* missense allele are viable without other obvious phenotypes. In support of the specificity of the *dz222* allele, it is worth noting that the exact same molecular change was identified by *Zou et al. (2018)* in their screen for genes involved in PVD patterning. The phenotype of *act-4(dz222)* is in stark contrast to the embryonic lethality observed upon continuous RNAi-mediated knockdown of *act-4* (*Gottschalk et al., 2005*, data not shown). Moreover, knock down of ACT-4/Actin by RNAi during larval stages on the one hand and, the G151E missense mutation in ACT-4/Actin encoded by the *act-4(dz222)* allele on the other hand result in similar phenotypes in PVD dendrites. In other words, loss of ACT-4 is as detrimental for dendrite patterning as the stronger biochemical interaction between TIAM-1 and ACT-4 (G151E). This suggests that the interaction between TIAM-1 and ACT-4, and possibly DMA-1/LRR-TM has to be transient or dynamic. Alternatively, the strong interaction between ACT-4(G151E) and TIAM-1/GEF could sequester actin from the polymerizing pool. These findings are reminiscent of a point mutation in β-spectrin, which by increasing the affinity of β-spectrin with actin one thousand fold, results in defects in Drosophila dendrite arborization neuron (*Avery et al., 2017*). Collectively, our studies and the findings by Zou et al. (*Zou et al., 2018*) suggest that at least two partially redundant pathways act downstream of the DMA-1/LRR-TM receptor. In one pathway, DMA-1 interacts via HPO-30/Claudin with the WRC during patterning of lower order branches, whereas the interaction with TIAM-1/GEF may directly link the receptor complex with the actin cytoskeleton during 4°

dendrite formation. Further experiments will be required to determine how these two pathways are spatially and temporally regulated.

### TIAM-1/GEF functions independently of its Rac1 GEF activity

Guanine nucleotide exchange factors such as TIAM-1 activate small GTPases by promoting the exchange of GDP for GTP. Tiam1 in vertebrates has been shown to serve important functions during nervous system patterning (*Yoo et al., 2012*). Knock down and overexpression experiments further established functions for Tiam1 in axon growth cone formation and, activity-dependent remodeling of dendritic arbors in vitro and in vivo (*Kunda et al., 2001*; *Tolias et al., 2005*; *Tolias et al., 2007*). Such functions may be conserved, since the drosophila homolog *still life/Tiam1* and *C. elegans TIAM-1/GEF* function during synaptic and, axonal patterning downstream of the netrin receptor UNC-40/DCC (Deleted in Colorectal Cancer), respectively (*Sone et al., 1997*; *Demarco et al., 2012*). All known functions of Tiam1 appear dependent on the canonical enzymatic activity of exchanging GDP for GTP in small GTPases of the Rac1 type.

Unexpectedly, our findings imply that TIAM-1/GEF can function independently of its Rac1 GEF activity in shaping PVD dendrites. This conclusion is supported by three arguments. First, transgenic expression of a TIAM-1 (T548F) fully rescued the *tiam-1* mutant phenotype. Second, engineering the T548F point mutation into the *tiam-1* locus resulted in animals with PVD dendritic arbors that were indistinguishable from wild type animals. The T548F mutation is analogous to the S1216F mutation in UNC-73/Trio, which lacks Rac1 activity, but displays neuronal patterning defects (*Steven et al., 1998*). Moreover, the very mutation in *C. elegans* TIAM-1/GEF also shows no Rac1 GEF activity in vitro (*Demarco et al., 2012*). Lastly, single and double mutants of the small GTPases *mig-2/RhoG* and *ced-10/Rac1*, previously established as acting downstream of TIAM-1/GEF during axonal patterning in *C. elegans* (*Demarco et al., 2012*), displayed no defects PVD morphology. Similarly, a third Rac-like gene (*rac-2/3*) encoded in the *C. elegans* genome serves no function individually in PVD patterning. Taken together, these findings suggest that TIAM-1/GEF can function independently of Rac1 activity during PVD patterning. To our knowledge, there is only one other known example of GEF-independent functions of a guanine nucleotide exchange factor. The DOCK 180 family member DOCK7 functions during nuclear migration by antagonizing TACC3, a protein known to coordinate microtubule polymerization (*Yang et al., 2012*). Thus, GEF-independent functions that directly modulate the cytoskeleton may be an additional and, previously underappreciated property of guanine nucleotide exchange factors.

## Materials and methods

### C. elegans strains

All strains were maintained using standard conditions (unless stated otherwise). For details see *Supplementary file 1*.

### Imaging and morphometric analyses

All strains were maintained using standard methods (*Brenner, 1974*). All experiments were performed at 20°C, and animals were scored as 1-day-old adults unless otherwise specified. The strains and mutant alleles used in this study are listed in the *Supplementary file 1*. Fluorescent images were captured in live *C. elegans* using a Plan-Apochromat 40×/1.4 or 63x/1.4 objective on a Zeiss Axioimager Z1 Apotome. Worms were immobilized using 5 mM levamisole and *Z* stacks were collected. Maximum intensity projections were used for further analysis.

For quantification of branching, synchronized starved L1 larvae were allowed to grow for 48 hr (*Figure 1*) or 50 hr (*Figure 2*) or, directly picked as L4 animals and mounted onto an agar pad for microscopy (*Figure 5*). Morphometric data was either obtained directly on the microscope (*Figure 1*) or, fluorescent images of immobilized animals (1–5 mM levamisol, Sigma) were captured using a Zeiss Axioimager equipped with an ApoTome. *Z* stacks were collected and maximum projections were used for counting dendritic branches manually (*Figure 5*) or by tracing dendrites as described (*Salzberg et al., 2013*) (*Figure 2*). In any case, all branches within 100 μm of the primary branch anterior to the cell body were counted and classified into primary, secondary, tertiary, and quaternary. All protrusions from the 1° dendrite were considered 2° branches.

Statistical comparisons were conducted using one-sided ANOVA with the Sidak or Tukey correction, the Kruskal-Wallis test, the Student t-test or the Mann-Whitney test as applicable using the (Prism 7 [GraphPad]) software suite.

## Quantification of fluorescence reporters

For quantification of UtrCH::tagRFP staining, images were acquired at a setting where maximal intensity of UtrCH::tagRFP staining was within the dynamic range of the camera detector. All end branches were then manually examined and classified as displaying staining or no staining if no discernible differences between the dendritic branch and surrounding background signals was detected, that is that no structure was visible in the UtrCH channel, even with maximum gain and contrast.

For quantification of DMA-1::GFP reporter membrane staining, animals were imaged with identical parameters to allow for intensity comparisons. The average fluorescence was then quantified using ImageJ. Briefly, the freehand line tool was used to trace the branches to create an ROI with a width of 10 pixels (=1.6 µm). For quantification on the primary dendrite, ROI lengths of 30, 40, 50 and 60 µm from the cell body were defined and the mean of the average fluorescent intensity of these 4 ROIs was calculated for each animal. For quantification on the tertiary branches, 4 ROIs were traced through the whole length of 4 separate branches, and the mean of the average fluorescence intensity of these 4 ROIs was calculated for each animal. As background control, four additional ROIs of the same length were defined in an area inside the worm but outside of branches, and means of the average fluorescent intensity of these ROI was calculated as the background for each animal and subtracted from the mean fluorescent intensity for either primary or tertiary branches, respectively. Statistical comparisons were conducted using one-sided ANOVA with the Tukey correction, the Kruskal-Wallis test, the Mann-Whitney test as applicable using the (Prism 7 [GraphPad]) software suite.

## Cloning of mutant alleles

Alleles of different genes were isolated during genetic screens for mutants with defects in PVD patterning. In several F1 clonal genetic screens for mutants with defects in PVD dendrite arborization, we isolated one allele of *dma-1* (*dz181*), two alleles of *hpo-30* (*dz178* and *dz189*), and two alleles of *tiam-1* (*dz184* and *dz206*). One allele of *act-4* (*dz222*) was isolated in an enhancer screen of the hypomorphic *kpc-1(gk333538)* allele. In addition, we obtained putative null deletion alleles for *dma-1(tm5156)*, *hpo-30(ok2047)*, and *tiam-1(tm1556) and tiam-1(ok772)*. Details for cloning of individual point mutations are provided below.

### tiam-1(dz184)

A SNP-mapping-WGS approach (*Doitsidou et al., 2010*) was used to map *tiam-1(dz184)* between 3 Mb and 6 Mb of chromosome I. Using SNP mapping (*Davis et al., 2005*), the mutation was further mapped to a physical interval between 4 Mb and 5.48 Mb. This region contained only one nonsense mutation. The mutation was confirmed by Sanger sequencing of the original isolate, confirming a non-sense C to T mutation in the *tiam-1(dz184)* allele. Additionally, the *tiam-1(dz184)* mutant showed non-complementation with all other loss-of-function alleles and its phenotype was not different from the *dz206*, *tm1556 and ok772* deletion alleles (see List of complementation tests below). Therefore, our genetic and molecular evidence suggests that all *tiam-1* alleles used in this study are likely complete loss-of-function alleles.

### tiam-1(dz206)

Complementation tests with a deletion allele of *tiam-1* showed non-complementation between *tiam-1(tm1556)* and *tiam-1(dz206)*. The mutation was further confirmed by Sanger sequencing of the original isolate, identifying a non-sense mutation C to T mutation in *dz206* in the *tiam-1* locus (see List of *tiam-1* mutant alleles below).

### hpo-30(dz178)

A SNP-mapping-WGS approach (*Doitsidou et al., 2010*) was used to map hpo-*30(dz178)* between 3 Mb and 5 Mb of chromosome V. This region contained only one candidate mutation. The mutation

was confirmed by Sanger sequencing of the original isolate, confirming a missense C to T mutation in *hpo-30(dz178)*. Additionally, the *hpo-30(dz178)* phenotype was not different from the *dz178 and ok2047* deletion allele. Therefore, our genetic and molecular evidence suggests that all *hpo-30* alleles used in this study are likely complete loss-of-function alleles.

### hpo-30(dz189)

Complementation tests with the other *loss-of-function* allele of *hpo-30* showed non-complementation between *hpo-30(dz178)* and *hpo-30(dz189)*. The mutation was further confirmed by Sanger sequencing of the original isolate, identifying a non-sense mutation C to T mutation in *hpo-30 (dz189)* (see List of *hpo-30* mutant alleles below).

### act-4(dz222)

The *dz222* allele was identified in a genetic screen as a recessive enhancer of the *kpc-1(gk333538)* hypomorphic allele. A SNP-mapping-WGS approach (*Doitsidou et al., 2010*) was used to map *act-4 (dz222)* between 4 Mb and 6 Mb of chromosome X. This region contained five candidate mutations, including non-sense, missense, frameshift and splice site mutations. The mutant defect in *dz222* was phenocopied by RNAi-mediated gene knock down of *act-4* and, rescued by transgenic expression of *act-4* under control of heterologous promoters. Sequencing of the original isolate identified a G to A missense mutation in *act-4(dz222)*, resulting in glycine to glutamic acid change at position 151 in ACT-4. Finally, when the *act-4(dz222)* allele was placed over a deficiency the phenotype was not enhanced. Therefore, our genetic and molecular evidence suggests that the *act-4(dz222)* allele used in this study is likely complete loss of function allele of *act-4* in the context of PVD dendrite morphogenesis.

### List of mutant *dma-1* alleles

| Allele | Molecular lesion | Defect in PVD | Physical Location[†] |
|---|---|---|---|
| *dz181* | splice acceptor | Yes | 8,283,604, G - > A |
| *tm5159* | deletion (671nt) | Yes | 8,282,281–8,282,951 |
| *dz266* | Y600Amber (ΔPDZ) | Yes | 8,284,283, A - > T[‡] |

[†]Nucleotide change according to nucleotide positions in Wormbase release WS260.
[‡]Note that additional synonymous changes were introduced as a result of the CRISPR procedure. See Repair oligos below for details.

### List of mutant *hpo-30* alleles

| Allele | Molecular lesion | Defect in PVD | Physical location[†] |
|---|---|---|---|
| *dz178* | S155F | Yes | 12,309,368 |
| *dz189* | Q168X | Yes | 12,309,330 |
| *ok2047* | Deletion (1,294 bp) | Yes | 12,309,724–12,311,017 |

[†]Nucleotide change according to nucleotide positions in Wormbase release WS252.

### List of mutant *tiam-1* alleles

| Allele | Molecular lesion | Defect in PVD | Physical location[†] |
|---|---|---|---|
| *dz184* | Q303X | Yes | 4,422,065 |
| *dz206* | Q543X | Yes | 4,424,113 |
| *dz264* | T548F | No | 4,424,128–4,424,129 |
| *dz265* | T548F | No | 4,424,128–4,424,129 |
| *tm1556* | Deletion (851 bp) | Yes | 4,423,487–4,424,337 |
| *ok772* | Deletion (838 bp) and insertion (18 bp) | Yes | 4,424,394–4,425,231 |

[†]Nucleotide change according to nucleotide positions in Wormbase release WS252.

## List of mutant *act-4* alleles

| Allele | Molecular lesion | Defect in PVD | Physical location[†] |
|--------|------------------|---------------|---------------------|
| dz222 | G151E (M03F4.2a) | Yes | 4,963,194 |
| gk279371 | L9H (M03F4.2a) | No | 4,960,687 |
| gk279385 | G246R (M03F4.2a) | No | 4,963,478 |
| gk473333 | A30T (M03F4.2a) | No | 4,962,830 |
| gk785720 | E4K (M03F4.2c) | No | 4,962,048 |

[†]Nucleotide change according to nucleotide positions in Wormbase release WS252.

## List of complementation tests for *tiam-1*

| Genotypes | % of defective PVDs | N | Results |
|-----------|---------------------|---|---------|
| +/+ (wdIs52 (control)) | 0 | 100 | Complementation |
| dz184/tm1556 | 100 | 100 | Non-complementation |
| dz184/ok772 | 100 | 100 | Non-complementation |
| dz184/+ | 0 | 100 | Complementation |
| dz206/tm1556 | 100 | 40 | Non-complementation |
| dz206/+ | 0 | 40 | Complementation |

## Molecular biology and transgenesis

To establish DNA constructs used for rescue experiments or immunoprecipitation experiments, the respective cDNAs were cloned under control of either cell-specific promoters or promoters to drive expression in cell culture. Details for plasmid construction can be found in *Supplementary file 1*.

## Transgenesis

A complete list of all transgenic strains created for this study is shown in the *Supplementary file 1*.

## PVD-specific rescue of *dma-1* PVD dendrite branching defect

Complete rescue of the *dma-1* mutant phenotype was achieved in transgenic animals carrying a plasmid where the *dma-1* cDNA was driven under control of a short version of the previously reported *ser-2prom3* (=*ser-2p3*) promoter (*Tsalik et al., 2003*). The new promoter, which we term *ser-2p3s* encompasses 1825 bp upstream of the predicted translational start site of the C02D4.2d isoform.

For the deletion of the intracellular domain in *dma-1* (ΔICD), a mCherry fluorescent tag was inserted in frame after E532 leaving a protein devoid of the last 71 residues. Both constructs were injected individually at 5 ng/µl into *dma-1(tm5159) I; wdIs52 II*, together with the *myo-3p::tagRFP* marker (at 50 ng/µl) for the injection mix containing the entire *dma-1* cDNA or *myo-2p::mCherry* (at 50 ng/µl) for the *dma-1* (ΔICD) version. The final injection mix was brought to 100 ng/µl of DNA concentration with *pBluescript*.

## Heterologous rescue of *tiam-1* PVD dendrite branching defect

The *tiam-1* cDNA was cloned under control of heterologous promoters: hypodermal *dpy-7p* (*Gilleard et al., 1997*), body wall muscle *myo-3p* (*Okkema et al., 1993*), pan-neuronal *rab-3p* (*Nonet et al., 1997*), and a PVD-specific *ser-2p3* promoter (*Altun-Gultekin et al., 2001*). All constructs were injected at 5 ng/µl into *tiam-1(tm1556) I; wdIs52 II*, together with the *myo-3p::mCherry* marker at 50 ng/µl and *pBluescript* at 50 ng/µl.

## Heterologous rescue of *act-4* PVD dendrite branching defect

The *act-4* (or *act-1*) genomic DNA was cloned under control of heterologous promoters: hypodermal body wall muscle *myo-3p* (*Okkema et al., 1993*) and PVD-specific *ser-2p3* (*Altun-Gultekin et al.,*

*2001*). All constructs were injected at 5 ng/µl into *wdIs52 II; act-4(dz222) X*, together with the *elt-2p::gfp* marker at 5 ng/µl and *pBluescript* at 90 ng/µl.

## Transcriptional reporter of *act-4*

The *act-4* transcriptional reporter promoter was constructed by cloning 2.6 kb upstream of the predicted *act-4* translational start site (based on *Figure 1—figure supplement 1*) into *pPD95.75-NLS::mCherry,* and injected at 5 ng/µl into *wdIs52 II* animals, together with the *myo-2p::mNG* marker at 5 ng/µl and *pBluescript* at 90 ng/µl.

## Translational reporter of *act-4*

The *act-4* cDNA was tagged at the C-terminus with tagRFP and expressed under the *Pser-2p3* promoter (*Okkema et al., 1993*; *Tsalik et al., 2003*). It was injected at 5 ng/µl into *wdIs52 II; act-4 (dz222)* animals together with the *myo-2p::mNG* marker at 10 ng/µl and *pBluescript* at 85 ng/µl.

## *tagRFP::UtrCH* and *tba-1::tagRFP* reporters

UtrCH and *tba-1* were tagged with tagRFP and expressed under *ser-2p3* promoter. All constructs were injected at 5 ng/µl into *wdIs52 II*, together with the *myo-3p::tagBFP* marker at 10 ng/µl and *pBluescript* at 85 ng/µl.

## Generation of genome-edited strains through CRISPR/Cas9

For introducing mutations into endogenous loci of *dma-1(Y600Amber)* and *tiam-1(T548F)*, the CRISPR co-conversion method as described in *Arribere et al. (2014)* was utilized. Briefly, two appropriate sgRNA sequences were identified for each gene and introduced into Cas9-sgRNA expression vector *pDD162*, while repair oligos were ordered from IDT. Both Cas9-sgRNA constructs and repair oligo were injected into N2 animals along with Cas9-sgRNA and a repair oligo for *dpy-10(cn64)* conversion. Rollers (*dpy-10(cn64)/+animals*) from the F1 generation were selected and genotyped using restriction sites engineered in the repair oligo. Wildtype F2 animals were then selected from positive F1 animals to segregate the intended edit from *dpy-10(cn64)*.

## sgRNA sequences

| dma-1(∆PDZ [Y600Amber]) | tiam-1 (T1548F) |
| --- | --- |
| ATGCCAAAATAGGATGATCC | AAGAATCATAAGAAGTAGTG |
| ATGCCAAAATAGGATGATCC | GATGGCTCTGCAAGAATTGT |

## Repair oligo sequences

*dma-1(∆PDZ[Y600Amber]):*
    TGCAAATAATCACACGGATTTAGATTTTTGAAACATTTGAAACTCTCAAAAGGATATTTATTATC
TAGACTAAGAGGAACCTGGCTTCGGAGGTGCTGGTGGAATCAATGGAGGAGTGGCTTTAAATG
TTTCAGTACTAACCAAGAATGG
    *tiam-1(T1548F):*
    TGTCAATAAGAAAACGGGCTGTGCAAAGTTGGCGATGGCTCTGCAAGAATTGTTAGTCTTTGA-
GAAGAAATATGTCAGCGATCTTCGAGAGGTAAGAGATCCCAAAAAGTTATAGAATTGAATAA
TTTACGATTTCAGATGA

## Cell culture, transient transfection and co-immunoprecipitation

HEK293T human embryonic kidney cells were maintained with a standard protocol in DMEM supplemented with 10%FBS. Cells were not validated because they were merely used for heterologous protein expression. The identity of heterologously expressed proteins was verified in western blots via unique immunotags. Transient transfection was performed using Lipofectamine 3000 (Thermofisher) according to the manufacturer's instruction. For each experimental permutation, $10^5$ cells/well were transfected in duplicate in six-well plates with 1 µg of each construct.

    For co-immunoprecipitation assays, the duplicate wells of $10^5$ transfected cells were lysed in 500 µl of lysis buffer and combined (25 mM Hepes, 150 NaCl, 1% Triton X-100, 0.5% Na-deoxycholate,

0.1% SDS, protease inhibitor cocktail (Thermo-Fisher)). The lysate was subsequently incubated at 4°C with agitation for an hour before centrifugation at 20,000 rcf for 10 min. The resulting lysate was combined with 20 µl of pre-equilibrated Protein A/G agarose (Thermo-fisher) and 1 µl of the appropriate antibody for pulldown and, incubated at 4°C with agitation overnight. The agarose was then washed 5 x with 500 µl of lysis buffer before SDS loading buffer was added. SDS-PAGE and western blot were then performed using standard protocols. All primary antibodies were used at 1:5000 dilutions and secondary antibodies were used at 1:10,000 dilutions. Supersignal West Femto ECL reagent (Thermo Fisher) was used for detection with X-ray films. For details on antibodies used see *Supplementary file 1*.

## Acknowledgements

We thank T Boulin, S Cook, A Meléndez, R Townley and members of the Bülow laboratory for comments on the manuscript and for discussion during the course of this work. We thank the Caenorhabditis Genome Center for strains (funded by NIH Office of Research Infrastructure Programs, P40OD010440); O Hobert and K Shen for reagents and discussions. This work was funded in part through the NIH (R01NS096772 to HEB; T32GM007288 and F31HD066967 to CADB; F31NS100370 to MR; T32GM07491 to MILP; P30HD071593 & P30CA013330 to Albert Einstein College of Medicine), a Human Genome Pilot Project from Albert Einstein College of Medicine and the Binational Science Foundation (#2013188 to HEB). NJRS is the recipient of a Fulbright-Colciencias fellowship. HEB is an Irma T. Hirschl/Monique Weill-Caulier research fellow.

## Additional information

### Funding

| Funder | Grant reference number | Author |
| --- | --- | --- |
| National Institute of General Medical Sciences | T32GM007288 | Carlos A Diaz-Balzac |
| Eunice Kennedy Shriver National Institute of Child Health and Human Development | F31HD066967 | Carlos A Diaz-Balzac |
| National Institute of Neurological Disorders and Stroke | F31NS100370 | Maisha Rahman |
| Fulbright-Colciencias | | Nelson J Ramirez-Suarez |
| National Institute of General Medical Sciences | T32GM07491 | Maria I Lázaro-Peña |
| Irma T Hirschl/Monique Weill-Caulier | | Hannes E Bülow |
| National Institute of Neurological Disorders and Stroke | R01NS096672 | Hannes E Bülow |

The funders had no role in study design, data collection and interpretation, or the decision to submit the work for publication.

### Author contributions

Leo TH Tang, Carlos A Diaz-Balzac, Conceptualization, Data curation, Formal analysis, Funding acquisition, Investigation, Visualization, Methodology, Writing—original draft, Writing—review and editing; Maisha Rahman, Nelson J Ramirez-Suarez, Data curation, Formal analysis, Investigation, Visualization, Writing—review and editing; Yehuda Salzberg, Data curation, Formal analysis, Investigation, Visualization; Maria I Lázaro-Peña, Investigation; Hannes E Bülow, Conceptualization, Resources, Data curation, Formal analysis, Supervision, Funding acquisition, Validation, Visualization, Project administration, Writing—review and editing

Author ORCIDs
Carlos A Diaz-Balzac (iD) https://orcid.org/0000-0002-4723-1282
Nelson J Ramirez-Suarez (iD) http://orcid.org/0000-0001-7394-860X
Hannes E Bülow (iD) http://orcid.org/0000-0002-6271-0572

Decision letter and Author response
Decision letter https://doi.org/10.7554/eLife.38949.020
Author response https://doi.org/10.7554/eLife.38949.021

## Additional files

### Supplementary files

• Supplementary file 1 Lists of all mutants alleles used, as well as all plasmids with details of their construction. In addition, *Supplementary file 1* lists all transgenic strains created during the course of these studies.
DOI: https://doi.org/10.7554/eLife.38949.023

• Transparent reporting form
DOI: https://doi.org/10.7554/eLife.38949.024

### Data availability

All data generated or analyzed during this study are included in the manuscript and supporting files.

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
