## [Decision Letter]

Thank you for submitting your article "TIAM-1/GEF can shape somatosensory dendrites independently of its GEF activity by regulating F-actin localization" for consideration by *eLife*. Your article has been reviewed by two peer reviewers, and the evaluation has been overseen by Shai Shaham, serving as guest Reviewing Editor, and Didier Stainier as the Senior Editor. The following individual involved in review of your submission has agreed to reveal his identity: Benjamin Podbilewicz (Reviewer #1).

The reviewers have discussed the reviews with one another and the Reviewing Editor has drafted this decision to help you prepare a revised submission.

The manuscript by Tang et al. describes new interactions between the cytoplasmic machinery and membrane receptors required for the patterning of complex but stereotypic dendritic trees of the PVD in *C. elegans*. Previous work by different labs has shown that an adhesion complex between DMA-1/LRR-TM in the PVD, MNR-1/Menorin and SAX-7/LICAM in the hypodermis and LECT-2/chemotaxin II from muscles collectively target and pattern dendritic trees of the PVD in *C. elegans*. The signaling pathways downstream of DMA-1 in the dendrites are studied in the current manuscript. Tang et al. show that HPO-30/claudin-like tetraspan receptor forms a complex with DMA-1/LRR-TM that is necessary for its localization and functions in PVD dendrites. In addition, they show that DMA-1, HPO-30, TIAM-1/GEF and ACT-4/Actin interact forming a complex. The authors suggest that this is a signaling complex and that it is necessary for the exclusion of microtubules from distal endings as well as for the localization of microfilaments in the terminal branches of the PVD. They propose that DMA-1 regulates F-actin localization and dynamics via TIAM-1 and that this function appears to be independent of its RAC-1 GEF activity.

The paper reports a set of interesting and timely findings. Nonetheless, the reviewers had concerns in three areas, which should be addressed in the revision:

1) While the epistasis studies are well done, a number of alternative possibilities in interpretation exist. At a minimum, these alternatives need to be given robust discussion in the revised paper. Better yet, some experiments to solidify some of the interpretations would be useful.

2) While the in vitro binding studies are supportive of the genetics, more controls to demonstrate direct relevance of binding in vivo are important. This could be accomplished by looking at localization of different proteins in the background of mutants (this was done in one case), or by looking at effects of a particular genetic mutant on binding (as was done in another case).

In addition, it is also important to demonstrate that the in vitro studies are reproducible – how many times was each study performed, and how variable were the results?

3) Given overall similarities with the recent paper from Zou et al., the authors should explicitly discuss differences/similarities and any complementary data the current paper provides.

Below are the specific comments raised by the two reviewers that can help to guide your revised submission:

Reviewer #1:

1) The genetic interactions and epistatic analyses should be simplified and alternative pathways may be discussed. For example, is it possible that the experiments can be interpreted as parallel pathways instead of a linear pathway as shown in Figure 3N?

2) The specificity of the biochemical interactions in vitro between DMA-1, TIAM-1 and ACT-4 should be validated in an in situ system. For example, colocalization at higher resolution of these proteins in the worm (e.g. by super resolution) could strengthen this part. I am concerned that some of the colocalization is in intracellular transport vesicles and not on the plasma membrane of the dendrites and cell body. Is it possible that any PDZ domain will interact with both actin and DMA-1? It would be important to see that these interactions also occur in worms and that they are specific. Live imaging can reveal whether the colocalized proteins are on the plasma membrane or alternatively that they colocalize in moving puncta (carrier vesicles).

3) The authors could use high-resolution methods (e.g. super resolution or FRET) to show that HPO-30 and DMA-1 colocalize (Figure 4). They could be in the same vesicular carriers. In contrast they may interact and colocalize at the plasma membrane of dendrites or cell body. To show direct in vivo interactions careful colocalization can be performed. Another result that is difficult to understand is why there is no colocalization in the cell body.

4) The authors may want to add a model the showing extracellular complexes that are proposed to be linked via DMA-1 to intracellular complexes. Is there any biochemical evidence showing that extracellular and intracellular complexes are linked by DMA-1.

Reviewer #2:

The authors rely on an extensive array of genetic tests to deduce roles for various effectors in dendrite outgrowth. These results are impressive and informative. However, the authors also need to re-evaluate these results based on the following considerations.

1) Results for mutant effects on 2º branches are confusing. Figure 2E shows equal numbers of 2º branches in WT vs. mutants of hpo-30, tiam-1 and act-4. Representative images (Figure 2A-D), however, appear to show fewer, and in the case of hpo-30, much shorter lateral branches than WT. Previous reports have documented fewer 2º branches for hpo-30 (O'Brien et al., 2017; Smith et al., 2013) and act-4 (Zou et al., 2018). What are criteria for scoring 2º branches, any lateral protrusion or must a bona fide 2º branch reach the sublateral nerve cord?

2) (Regarding 2º branches). "Double mutants between sax-7/L1CAM and hpo-30/Claudin were not more severe than the more severe of the single mutants, indicating that hpo-30/Claudin functions in the menorin pathway." This statement is confusing and subject to alternative explanations. Why not just say that the double mutant shows the hpo-30 phenotype (i.e., hpo-30 is epistatic) and that this means that sax-7 and hpo-30 both regulate lateral branching. Formally, hpo-30 could function either downstream of sax-7 for dendrite outgrowth or upstream since lateral branching, which hpo-30 promotes in the WT, is certainly necessary for expression of the sax-7 mutant hyperbranching phenotype.

3) (Subsection “The hpo-30/Claudin, tiam-1/GEF and act-4/Actin genes act in the menorin pathway”, first paragraph). Genetic evidence presented here for parallel roles for hpo-30 and sax-7 in higher order branching is not convincing. This finding is based on a double mutant which, according to authors, shows an enhanced 3º branch phenotype. It's hard to imagine how 3º branches could be scored accurately given the highly disordered appearance of PVD menorahs in sax-7 mutants (Dong et al., 2013; Salzberg et al., 2013). It should be sufficient to cite the finding that the hpo-30 lateral branching phenotype is more severe than that of sax-7 as evidence of an independent role for hpo-30.

4) (Subsection “The hpo-30/Claudin, tiam-1/GEF and act-4/Actin genes act in the menorin pathway”, first paragraph). Genetic results showing that dma-1 is epistatic to both hpo-30 and tiam-1 do indicate, as noted by the authors, that dma-1 is required for higher order branching. But why are results for tiam-1, act-4 double mutants also reported in this sentence? In fact, Figure 2 shows a much stronger 2º branch defect in tiam-1, act-4 doubles than for either single mutant which points to parallel roles in branching (see comment about null alleles and below). The authors do not comment on this point.

5) Comment on double mutant analysis (Figure 2). Are these null alleles? If not, then binary combinations of hypomorphic alleles that result in more severe phenotypes cannot distinguish between genes that function in common vs. parallel pathways.

6) (Subsection “The hpo-30/Claudin, tiam-1/GEF and act-4/Actin genes act in the menorin pathway”, second paragraph). Suppression of baobab tree phenotype is also subject to an alternative interpretation. MNR-1 is ectopically expressed in muscle to produce a PVD hyperbranching effect but this effect requires secondary branches to grow out to the sublateral location (body muscles) of the ectopic MNR-1 cue. This rarely happens in hpo-30 mutants, for example, and thus could account for hpo-30 epistasis in this case.

Additional comments:

(Subsection “HPO-30/Claudin functions to localize DMA-1/LRR-TM”). (Figure 4A-C) Images of hpo-30 and dma-1 co-localization are not convincing. Higher resolution images are needed particularly of lateral branches where hpo-30 and dma-1 are proposed to function. For example, why not include high mag insets for the boxes shown in Figure 4B? How are puncta defined? Is the more severe branching phenotype for dma-1 than hpo-30 consistent with the observation here that hpo-30 is required for punctate localization of dma-1?

Comment about biochemical results (Figure 4, 7). These results provide extensive and convincing evidence of physical interactions among key components of the dendrite branching apparatus but results are not quantified. Why? Were these results replicated (n > 1)?

---

## [Author Response]

The paper reports a set of interesting and timely findings. Nonetheless, the reviewers had concerns in three areas, which should be addressed in the revision:

We have performed a large number of additional experiments, which have clarified certain questions, but also extended prior findings. These new results have resulted in a slight reorganization of the overall paper, in particular with regard to Figures 4, 5, 6 and 7 to allow for a more logical flow.

1) While the epistasis studies are well done, a number of alternative possibilities in interpretation exist. At a minimum, these alternatives need to be given robust discussion in the revised paper. Better yet, some experiments to solidify some of the interpretations would be useful.

We appreciate this comment and have revised the genetic interpretation to allow for alternative interpretations. In addition, we include qualifying sentences where necessary. We have also removed Figure 3N, because the genetic pathway shown in this Figure was too simplistic and thus potentially misleading. Finally, we have added some additional data, including e.g. a *dma-1; act-4* double mutant, which shows that pathways in addition to *dma-1* must exist and function in parallel. For more details see also individual responses to reviewers below.

2) While the in vitro binding studies are supportive of the genetics, more controls to demonstrate direct relevance of binding in vivo are important. This could be accomplished by looking at localization of different proteins in the background of mutants (this was done in one case), or by looking at effects of a particular genetic mutant on binding (as was done in another case).In addition, it is also important to demonstrate that the in vitro studies are reproducible – how many times was each study performed, and how variable were the results?

We have performed a number of experiments along these lines as suggested. These experiments have not only clarified certain aspects, but also resulted in new and important information:

1) We now show that TIAM-1::GFP localization is dependent on the last four amino acids of DMA-1 in vivo. While the paper from the Shen lab showed a dependence on *dma-1*, our specific PDZ deletion allele shows unequivocally that it is the PDZ ligand motif of DMA-1 that is required for TIAM-1::GFP localization in vivo. These data are now presented as Figure 5I.

2) We created a new HPO-30::tagBFP reporter to analyze potential colocalization between HPO30 and DMA-1 in vivo. While the intended FRET experiments were unsuccessful, we discovered that the amount of DMA-1::GFP on the membrane is HPO-30-dependent.

Specifically, we found using both gain and loss of function approaches that HPO-30 negatively regulates the membrane fraction of DMA-1::GFP. In addition, through time lapse analyzes, we can show that hpo-30 clearly also functions to control the number of mobile vesicles, that is trafficking of DMA-1::GFP in the cell. All of these data are now part of a completely revised/new Figure 4.

3) We have conducted additional binding experiments to determine the specificity of the PDZ ligand motif of DMA-1. Specifically, we have found that replacing the PDZ ligand motif of DMA1 with the PDZ motif of the heparan sulfate proteoglycan SDN-1/Syndecan has the same effect as removing the DMA-1 PDZ ligand motif entirely. These experiments establish that the interaction between the TIAM-1 PDZ domain and the DMA-1 ligand motif are specific, but suggest (as we had previously concluded) that other parts of DMA-1 must also play a role in binding. These data are now included as a new Figure 5—figure supplement 2A.

3) Given overall similarities with the recent paper from Zou et al., the authors should explicitly discuss differences/similarities and any complementary data the current paper provides.

We now more clearly reference the similarities between the Zou paper and ours, as well as where our paper extends findings from Zou et al., 2018.

Below are the specific comments raised by the two reviewers that can help to guide your revised submission:Reviewer #1:1) The genetic interactions and epistatic analyses should be simplified and alternative pathways may be discussed. For example, is it possible that the experiments can be interpreted as parallel pathways instead of a linear pathway as shown in Figure 3N?

We appreciate this comment and have revised the manuscript accordingly. First, we have removed the pathway shown in Figure 3N, because it was overly simplistic and did not take into account alternative explanations such as a pathway that may act in parallel. Second, we have now included alternative explanation as part of the Results section where applicable.

2) The specificity of the biochemical interactions in vitro between DMA-1, TIAM-1 and ACT-4 should be validated in an in situ system. For example, colocalization at higher resolution of these proteins in the worm (e.g. by super resolution) could strengthen this part. I am concerned that some of the colocalization is in intracellular transport vesicles and not on the plasma membrane of the dendrites and cell body. Is it possible that any PDZ domain will interact with both actin and DMA-1? It would be important to see that these interactions also occur in worms and that they are specific. Live imaging can reveal whether the colocalized proteins are on the plasma membrane or alternatively that they colocalize in moving puncta (carrier vesicles).

The reviewer makes a number of thoughtful suggestions, most of which we have addressed experimentally.

1) We analyzed TIAM-1::GFP localization in vivo and found that is critically dependent on the PDZ motif of DMA-1. This was accomplished by evaluating TIAM-1::GFP in a DMA-1 mutant, in which we selectively removed the last four amino acids (the PDZ ligand motif). These data are now included as Figure 5I.

2) The reviewer mentions that some of the colocalization of proteins may be in vesicles. This is most likely the case. In our revised version we have separately quantified vesicular and membrane localization of DMA-1::GFP where applicable and possible. These studies have actually revealed additional insight with regard to the interactions between DMA-1 and HPO30 (see below).

3) The reviewer raises the question of specificity for the PDZ domain interactions. We addressed this question by (a) testing TIAM-1::GFP localization in a DMA-1(δ PDZ) mutant (see above response to point 1). In addition, we have addressed specificity in vitro. Specifically, we have replaced the PDZ ligand motif in DMA-1 with the PDZ ligand motif of the heparan sulfate proteoglycan SDN-1/Syndecan. We can show that changing the last four amino acids of DMA1 reduces the TIAM-1/DMA-1 interaction in Co-IPs to the same extent as removing the PDZ ligand motif in DMA-1 completely. These findings suggest that the TIAM-1/DMA-1 interaction via the PDZ ligand motif and the PDZ binding domain is specific. These data are now included at Figure 5—figure supplement 2A.

4) The reviewer suggests live imaging and separation into vesicular and plasma membrane fractions. We have completed those experiment and made a number of significant additional observations. See response to point 3 below.

3) The authors could use high resolution methods (e.g. super resolution or FRET) to show that HPO-30 and DMA-1 colocalize (Figure 4). They could be in the same vesicular carriers. In contrast they may interact and colocalize at the plasma membrane of dendrites or cell body. To show direct in vivo interactions careful colocalization can be performed. Another result that is difficult to understand is why there is no colocalization in the cell body.

Following the reviewer’s suggestion, we created a new translational reporter for *hpo-30* fused to *tagBFP* in the hope to conduct FRET experiments between GFP-tagged DMA-1 and BFP-tagged HPO-30. Unfortunately, the FRET experiments were unsuccessful for unknown reasons. However, the double-labeled strain nonetheless allowed for much clearer colocalization and allowed several additional insights, including the finding that DMA-1 plasma membrane localization depends on the level of HPO-30/Claudin, suggesting a role for this protein in endocytosis and trafficking of DMA-1. These new findings are now included as Figure 4H-L.

4) The authors may want to add a model showing the extracellular complexes that are proposed to be linked via DMA-1 to intracellular complexes. Is there any biochemical evidence showing that extracellular and intracellular complexes are linked by DMA-1.

Following the reviewer’s suggestion, we have conducted co-transfections experiments, where we transfected TIAM-1 and ACT-4 in the presence and absence of the leucine rich transmembrane receptor DMA-1. These experiments suggest that the interaction between TIAM-1 and ACT-4 is stronger in the presence of DMA-1, thereby making a case for the PVD-specific signaling complex. These new data are now included as Figure 7F.

Reviewer #2:The authors rely on an extensive array of genetic tests to deduce roles for various effectors in dendrite outgrowth. These results are impressive and informative. However, the authors also need to re-evaluate these results based on the following considerations.1) Results for mutant effects on 2º branches are confusing. Figure 2E shows equal numbers of 2º branches in WT vs. mutants of hpo-30, tiam-1 and act-4. Representative images (Figure 2A-D), however, appear to show fewer, and in the case of hpo-30, much shorter lateral branches than WT. Previous reports have documented fewer 2º branches for hpo-30 (O'Brien et al., 2017; Smith et al., 2013) and act-4 (Zou et al., 2018). What are criteria for scoring 2º branches, any lateral protrusion or must a bona fide 2º branch reach the sublateral nerve cord?

The reviewer raises important points regarding slight discrepancies between our data and published data, which can also be observed in the published record, both in studies from different labs, but also the same lab. For example, as the reviewer mentions, O’Brien and Smith note less 2º branches in hpo-30 mutants, but with different absolute numbers. Smith et al., 2013 find an average of around 20 2º branches per hpo-30 mutant animal, whereas O’Brien et al., 2017 find 25 (or more) 2º branches on average per hpo-30 mutant animal. In contrast, Zou et al. and we see more (although at least in our case the difference is not significant) 2º branches in hpo-30 animals. There may be at least three reasons for these slight discrepancies between and within labs.

1) The definition of what constitutes a 2º branch (as the reviewer suggests). O’Brien and Smith et al., 2013 score all 2º along the whole animal. In contrast, we score any protrusion from the 1º in a 100µm segment anterior to the cell body as a 2º branch as does Zou et al., 2018, although it is not clear what constitutes a 2º branch in their case. Interestingly, Zou et al., 2018 also find a slight (but not statistically investigated) increase for 2º dendrites for hpo-30 mutants, just like we do, but in contrast to O’Brien et al. and Smith et al. This is most likely because of the difference in scoring criteria. A possible biological explanation could be a role for of hpo-30 in trafficking that we discovered, which may affect branches more distal from the cell body more than branches closer to the cell body.

2) Statistics. For example, we find a decrease for 2º branches in act-4 mutants, which is not significant in our case, but is for Zou et al., 2018. A possible explanation is that in our data set we make over 170 statistical comparisons (between the genotypes in Figure 2 and Figure 2—figure supplement 2), which requires substantial corrections for multiple comparisons. While our analyses may thus miss smaller differences, we can be fairly confident in any larger differences we observe. All conclusions we make in the paper are based on highly significant differences.

3) Systematic error. In our experiments we find that between experimenters, there can be slight differences, for example due to slight differences in staging or scoring. This phenomenon is, however, not limited to our lab, but also apparent in the published literature (see for example the slight differences between O’Brien et al., 2017 and Smith et al., 2013 mentioned above). We believe that as long as the same experimenter completes an experiment, the systematic error should be consistent and so should be the conclusions. In our revised version, the complete morphometric data from Figure 1, Figure 2, and Figure 5 was obtained by different individuals, respectively. While there are slight differences in absolute numbers for different branches the conclusions we draw are unambiguous.

In conclusion, to address the issues mentioned above as much as possible, we have (1) only data from one experimenter per experiment/figure and (2) describe as precisely as possible in the Materials and methods sections how the animals were staged and scored.

2) (Regarding 2º branches). "Double mutants between sax-7/L1CAM and hpo-30/Claudin were not more severe than the more severe of the single mutants, indicating that hpo-30/Claudin functions in the menorin pathway." This statement is confusing and subject to alternative explanations. Why not just say that the double mutant shows the hpo-30 phenotype (i.e., hpo-30 is epistatic) and that this means that sax-7 and hpo-30 both regulate lateral branching. Formally, hpo-30 could function either downstream of sax-7 for dendrite outgrowth or upstream since lateral branching, which hpo-30 promotes in the WT, is certainly necessary for expression of the sax-7 mutant hyperbranching phenotype.

With our scoring criteria, there is no difference with regard to the number of 2º branches between sax7 and hpo-30 mutants. Regardless, we have changed the conclusion to read more precisely “…indicating that *hpo-30/Claudin* functions in a pathway with *sax-7/L1CAM* for 2º branch patterning…”. The reviewer is correct that hpo-30 could be acting upstream or downstream of sax-7.

To not insinuate otherwise, we have removed the genetic pathway in Figure 3N.

3) (Subsection “The hpo-30/Claudin, tiam-1/GEF and act-4/Actin genes act in the menorin pathway”, first paragraph). Genetic evidence presented here for parallel roles for hpo-30 and sax-7 in higher order branching is not convincing. This finding is based on a double mutant which, according to authors, shows an enhanced 3º branch phenotype. It's hard to imagine how 3º branches could be scored accurately given the highly disordered appearance of PVD menorahs in sax-7 mutants (Dong et al., 2013; Salzberg et al., 2013). It should be sufficient to cite the finding that the hpo-30 lateral branching phenotype is more severe than that of sax-7 as evidence of an independent role for hpo-30.

Work from us and others (e.g. Dong et al., 2013; Salzberg et al., 2013; Zou et al., 2016; Diaz-Balzac et al., 2016) has shown that 3º branches can be scored reliably in mutants of sax-7, mnr-1, lect-2, and to some extent dma-1 and, that meaningful conclusion can be drawn. The difference in Figure 2F between a sax-7; hpo-30 double and the sax-7 and hpo-30 single mutants, respectively, is not subtle. In the double mutant there are essentially no 3º branches left. We thus prefer to leave our statement as is.

4) (Subsection “The hpo-30/Claudin, tiam-1/GEF and act-4/Actin genes act in the menorin pathway”, first paragraph). Genetic results showing that dma-1 is epistatic to both hpo-30 and tiam-1 do indicate, as noted by the authors, that dma-1 is required for higher order branching. But why are results for tiam-1, act-4 double mutants also reported in this sentence? In fact, Figure 2 shows a much stronger 2º branch defect in tiam-1, act-4 doubles than for either single mutant which points to parallel roles in branching (see comment about null alleles and below). The authors do not comment on this point.

We agree that mentioning the tiam-1; act-4 double in this sentence was not intuitive and we have accordingly removed the reference to this double mutant to now read “The *dma-1/LRR-TM; hpo30/Claudin* and *dma-1/LRR-TM; tiam-1/GEF* were statistically indistinguishable from the *dma-1/LRRTM* single mutant, suggesting that *dma-1/LRR-TM* is epistatic and required for most if not all functions during patterning of higher order branches in PVD.”

Instead, we now mention the tiam-1; act-4 double mutant in the following sentence “Interestingly, double mutants between *tiam-1/GEF* and *mnr-1/Menorin, lect-2/Chondromodulin II, kpc-1/Furin, sax7/L1CAM, hpo-30/Claudin*, or *act-4*, respectively,appeared more severe than either of the single mutants alone, but indistinguishable from the *dma-1/LRR-TM* single mutant (Figure 2E, Figure 2—figure supplement 2). These findings suggest that *tiam-1/GEF* also serves in a genetic pathway that functions in parallel to *mnr-1/sax-7/lect-2/hpo-30* and, possibly, *act-4*.”

5) Comment on double mutant analysis (Figure 2). Are these null alleles? If not, then binary combinations of hypomorphic alleles that result in more severe phenotypes cannot distinguish between genes that function in common vs. parallel pathways.

Thank you pointing this out. With the exception of act-4 all other alleles are molecular null alleles (either premature nonsense mutations or deletions). For act-4, we placed the dz222 allele over a deficiency that uncovers this region and found no enhancement of the PVD phenotype in the transheterozygous animal compared to the *dz222* homozygous mutant, suggesting that *act-4(dz222)* behaves as a genetic null allele for the observed PVD-related phenotypes. We now mention this fact in the legend to Figure 2 and refer to the supplemental material, where all these data are described in detail.

6) (Subsection “The hpo-30/Claudin, tiam-1/GEF and act-4/Actin genes act in the menorin pathway”, second paragraph). Suppression of baobab tree phenotype is also subject to an alternative interpretation. MNR-1 is ectopically expressed in muscle to produce a PVD hyperbranching effect but this effect requires secondary branches to grow out to the sublateral location (body muscles) of the ectopic MNR-1 cue. This rarely happens in hpo-30 mutants, for example, and thus could account for hpo-30 epistasis in this case.

This alternative explanation appears quite unlikely, because, while reduced in number, there are secondary branches in *hpo-30* mutants that reach the sublateral line where body wall muscle and the lateral hypodermis abut. The same is true for the *act-4* mutant allele and for *mnr-1/*Menorin mutants. Thus, for the alternative explanation to be plausible one would expect at least partial suppression of the baobab gain of function phenotype. Instead, we always see complete epistasis for *hpo-30, act-4* or *mnr-1* mutants (or any other mutant we have tested in the past), even if in these mutants a significant number of dendrites reached the boundary between the muscle and epidermis, where 3º dendrites are normally located.

Additional comments:(Subsection “HPO-30/Claudin functions to localize DMA-1/LRR-TM”). (Figure 4A-C) Images of hpo-30 and dma-1 co-localization are not convincing. Higher resolution images are needed particularly of lateral branches where hpo-30 and dma-1 are proposed to function. For example, why not include high mag insets for the boxes shown in Figure 4B? How are puncta defined? Is the more severe branching phenotype for dma-1 than hpo-30 consistent with the observation here that hpo-30 is required for punctate localization of dma-1?

We have created a new translational reporter for *hpo-30* fused to *tagBFP* and analyzed co-localization with DMA-1::GFP. This strain allowed for much clearer colocalization. We also refined our protein localization for DMA-1::GFP and separately quantified presumptive membrane localization and vesicular intracellular localization. These experiments provided several additional insights, including the finding that DMA-1 plasma membrane localization depended on the level of HPO-30/Claudin, suggesting a role for this protein in endocytosis and trafficking of DMA-1. These new findings are now included as Figure 4H-L.

Comment about biochemical results (Figure 4, 7). These results provide extensive and convincing evidence of physical interactions among key components of the dendrite branching apparatus but results are not quantified. Why? Were these results replicated (n > 1)?

Co-immunoprecipitation experiments are only semi-quantitative. However, we consider the results where we draw (cautious) semi-quantitative conclusions quite clear (see for example the binding between TIAM-1 with ACT-4 and ACT-4(G151E), respectively. All biochemical assays had always been performed at least 3 times. We now mention this fact explicitly in all applicable figure legends.